# Selective cell death in HIV-1-infected cells by DDX3 inhibitors leads to depletion of the inducible reservoir

Shringar Rao[1], Cynthia Lungu[2], Raquel Crespo[1], Thijs H. Steijaert [1], Alicja Gorska[1], Robert-Jan Palstra [1], Henrieke A. B. Prins [3], Wilfred van Ijcken [4], Yvonne M. Mueller [5], Jeroen J. A. van Kampen[2], Annelies Verbon[3], Peter D. Katsikis [5], Charles A. B. Boucher[2], Casper Rokx [3], Rob A. Gruters [2] & Tokameh Mahmoudi [1,6,7 ✉]

An innovative approach to eliminate HIV-1-infected cells emerging out of latency, the major hurdle to HIV-1 cure, is to pharmacologically reactivate viral expression and concomitantly trigger intracellular pro-apoptotic pathways in order to selectively induce cell death (ICD) of infected cells, without reliance on the extracellular immune system. In this work, we demonstrate the effect of DDX3 inhibitors on selectively inducing cell death in latent HIV-1-infected cell lines, primary CD4+ T cells and in CD4+ T cells from cART-suppressed people living with HIV-1 (PLWHIV). We used single-cell FISH-Flow technology to characterise the contribution of viral RNA to inducing cell death. The pharmacological targeting of DDX3 induced HIV-1 RNA expression, resulting in phosphorylation of IRF3 and upregulation of IFNβ. DDX3 inhibition also resulted in the downregulation of BIRC5, critical to cell survival during HIV-1 infection, and selectively induced apoptosis in viral RNA-expressing CD4+ T cells but not bystander cells. DDX3 inhibitor treatment of CD4+ T cells from PLWHIV resulted in an approximately 50% reduction of the inducible latent HIV-1 reservoir by quantitation of HIV-1 RNA, by FISH-Flow, RT-qPCR and TILDA. This study provides proof of concept for pharmacological reversal of latency coupled to induction of apoptosis towards the elimination of the inducible reservoir.

[1] Department of Biochemistry, Erasmus University Medical Center, Rotterdam, The Netherlands. [2] Department of Viroscience, Erasmus University Medical Center, Rotterdam, The Netherlands. [3] Department of Internal Medicine, Section of Infectious Diseases, Erasmus University Medical Center, Rotterdam, The Netherlands. [4] Erasmus Centre for Biomics, Erasmus University Medical Center, Rotterdam, The Netherlands. [5] Department of Immunology, Erasmus University Medical Center, Rotterdam, The Netherlands. [6] Department of Pathology, Erasmus University Medical Center, Rotterdam, The Netherlands. [7] Department of Urology, Erasmus University Medical Center, Rotterdam, The Netherlands. ✉email: t.mahmoudi@erasmusmc.nl

The main obstacle towards a human immunodeficiency virus type-1 (HIV-1) cure is the presence of a latent viral reservoir that does not actively produce viral particles, but retains the ability to do so and is not eliminated by combination antiretroviral therapy (cART)[1,2]. One strategy to eliminate the latent HIV-1 reservoir is the shock and kill approach that involves the use of latency-reversing agents (LRAs) to reactivate viral gene expression (Shock), followed by the elimination of cells harbouring reactivated provirus (Kill)[3]. The 'Kill' aspect of this approach has largely depended on an effective host adaptive immune response. Although clinical studies with LRAs demonstrated induced reactivation in vivo, limited to no reduction in the size of the viral reservoir or time to viral rebound post-cessation of cART was observed, indicating that elimination of reactivating cells warrants more attention[4–6]. So far, broadly neutralising antibodies, therapeutic vaccines, chimeric antigen receptors, checkpoint inhibitors and engineered bispecific antibodies have been investigated to eliminate reactivated HIV-1-infected cells (reviewed in[7]). However, dependency on cell-mediated and humoral immune responses, which are already compromised in HIV-infected individuals because of impaired CD4+ T cell help and exhausted immune compartments, pose a considerable obstacle to these strategies. Indeed, early results from combinatorial clinical trials which combine LRAs and immune-enhancing clearance strategies demonstrated that although HIV-1-specific immune responses were induced, no decrease in the size of the viral reservoir or viral control in the absence of cART were observed (refs. [8–10], reviewed in ref. [11]).

A parallel approach to eliminate the viral reservoir independently of the extracellular immune system is to pharmacologically trigger pathways that intrinsically induce cell death (ICD) in reactivating HIV-1-infected reservoir cells. For example, using compounds to activate innate intracellular signalling pathways could trigger apoptosis. Several HIV-1 proteins have been demonstrated to have both anti-apoptotic activities in the early stages of the viral replication cycle and pro-apoptotic activity in the later replication stages[12]. By inhibiting viral anti-apoptotic activity or by promoting pro-apoptotic functions, HIV-1-infected cells can be selectively eliminated, leading to a reduction in the size of the viral reservoir. Multiple pro-apoptotic compounds that have been developed in the context of cancer research could potentially be repurposed for viral reservoir elimination and some have been investigated for their ability to ICD of HIV-1-infected cells (reviewed in refs. [13,14]). Interestingly, an important characteristic of the latent viral reservoir that has recently come to light is that while it is transcriptionally competent, it does not necessarily generate viral proteins due to post-transcriptional blocks[15,16]. Induced viral RNA does, however, retain the ability to trigger innate antiviral signalling pathways[17,18] and possibly ICD in reactivating cells. The potential contribution of inducing viral transcription as a strategy to induce death of the infected cells has been thus far under-investigated[19].

Advanced single-cell fluorescence in situ hybridisation Flow cytometry (FISH-Flow)[15,16,20,21] provides a powerful tool to examine the specific contribution of HIV-1 unspliced RNA (herein referred to as vRNA) in inducing cell death. Unlike previous single-cell HIV-1 flow cytometry-based studies that could detect only viral protein production and the translationally competent reservoir, FISH-Flow allows for the delineation of the contribution of the under characterised transcriptionally competent cells that produce vRNA, can trigger innate immune responses and may or may not express viral proteins[19,22]. The single-cell analysis thus provides an additional layer of insight when studying the contribution of vRNA to apoptosis.

DEAD-box polypeptide 3, X-Linked, herein referred to as DDX3, is a host protein belonging to the DEAD-box (Asp-Glu-Ala-Asp) family of ATP-dependent RNA helicases. It is involved in all aspects of RNA metabolism including processing, transport and translation, as well as in cell cycle progression, stress response, innate immune sensing and apoptosis[23–25]. DDX3 has been implicated in several cancers due to its oncogenic function but also plays a dual role in cancer progression because of its tumour suppressive activity[26–30]. Moreover, DDX3 plays distinct roles in HIV-1 infection by enhancing viral RNA gene expression, but also in activating components of the innate antiviral signalling pathway[31]. During HIV-1 replication, DDX3 has an integral function in the nucleocytoplasmic export of the vRNA[32,33] as well as in vRNA translation in the cytoplasm[34–36]. In dendritic cells, DDX3 senses abortive HIV-1 RNA transcripts to induce type I interferon immune responses via mitochondrial antiviral signalling protein (MAVS)[37]. Because of its widely described roles in viral replication as well as in tumorigenesis, DDX3 has thus emerged as an attractive target for both anti-cancer and antiviral drugs[27,38].

Due to DDX3's roles in both HIV-1 RNA metabolism as well as in the regulation of apoptosis, we hypothesised that treatment of HIV-1-infected cells with small molecule inhibitors of DDX3 would inhibit vRNA export and translation; while the residual vRNA would activate innate antiviral immune signalling pathways and trigger apoptosis of vRNA-expressing cells, thereby resulting in a depletion of the viral reservoir.

Here we evaluated the effects of DDX3 inhibition on the latent HIV-1 reservoir using two clinically advanced inhibitors of DDX3, RK-33 and FH-1321, which target the ATPase and RNA-binding domains of DDX3, respectively. Pharmacological inhibition of DDX3 reversed HIV-1 latency in the latent HIV-1-infected J-Lat 11.1 cell line, ex vivo latently infected CD4+ T cells, and CD4+ T cells obtained from cART-suppressed HIV-1-infected donors, likely via NF-κB activation. DDX3 inhibitor-mediated HIV-1 latency reversal was observed predominantly at the RNA level, which is consistent with the role of DDX3 as an inhibitor of vRNA export and translation. Chemical inhibition of DDX3 resulted in induction of IRF3 phosphorylation, upregulation of IFNβ expression and apoptosis in vRNA-expressing cells. In accordance with its pharmacological inhibition, depletion of DDX3 via shRNA-mediated knockdown in J-Lat cells also reversed HIV-1 latency and induced IFNβ expression. RNA-sequencing analysis of primary CD4+ T cells obtained from uninfected independent donors revealed significant down-regulation of the inhibitor of apoptosis protein BIRC5 and Heat Shock protein 70 (HSP70) upon DDX3 inhibition. Importantly, treatment of latent HIV-1-infected cells with DDX3 inhibitors resulted in selective induction of apoptosis in vRNA-expressing cells but not in the uninfected/bystander cells. We quantitated the impact of treatment with DDX3 inhibitors on the inducible latent HIV-1 reservoir in CD4+ T cells obtained from cART-suppressed people living with HIV-1 (PLWHIV) in an in vitro culture model using three different methods of reservoir quantitation. Strikingly, treatment with DDX3 inhibitors over 5 days resulted in an ~50% reduction of the inducible reservoir as determined by quantitation of cell-associated vRNA, by tat/rev-induced limiting dilution assay, as well as at the single-cell level using FISH-Flow technology. These results demonstrate that pharmacological inhibition of DDX3 induces latency reversal and selective apoptosis of latent HIV-1-infected cells and serve as a proof-of-concept that ICD inducers can decrease the inducible HIV-1 reservoir in infected patient cells ex vivo.

## Results

**DDX3 inhibition induces viral reactivation in J-Lat 11.1 cells.** Our initial screening studies focus on examining whether

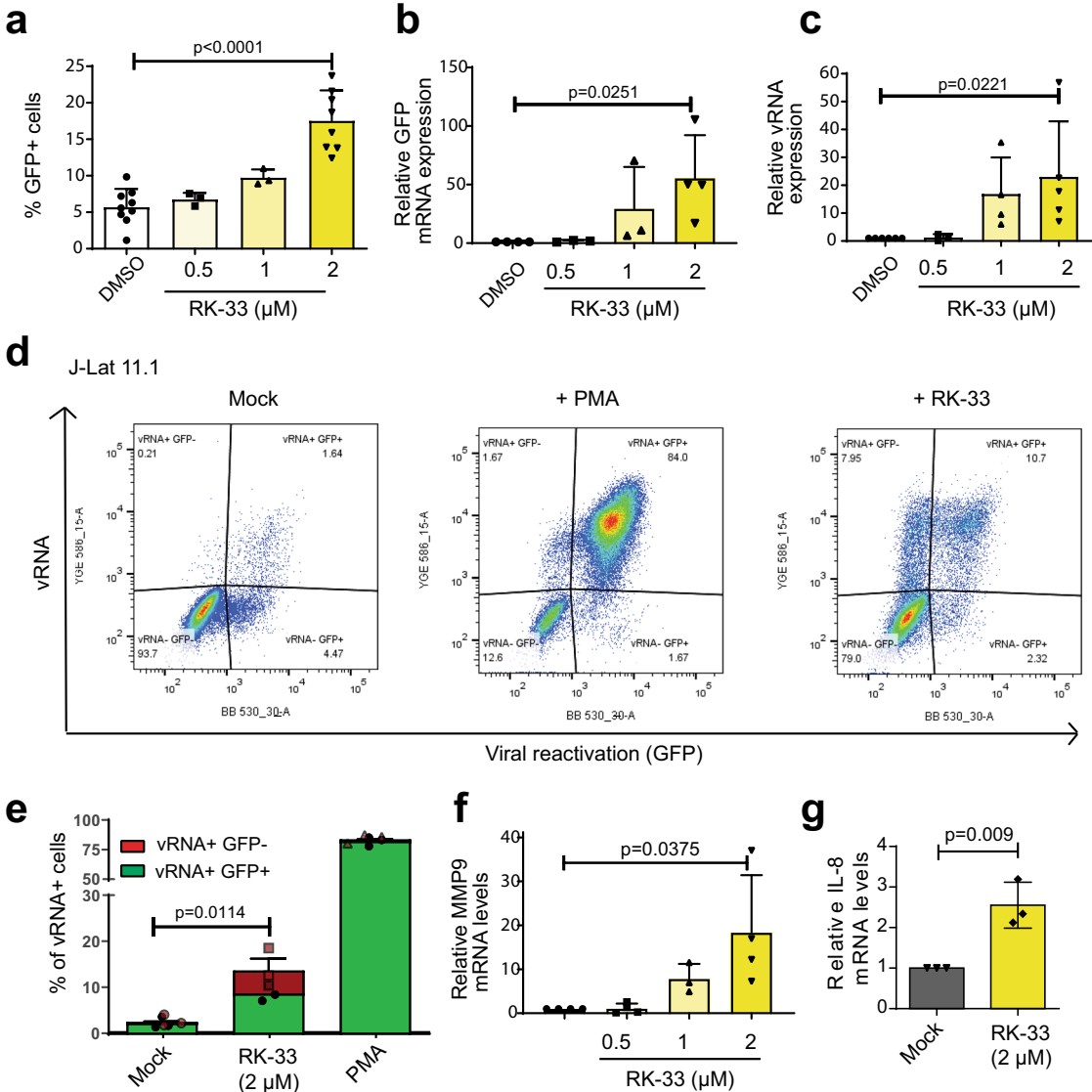

**Fig. 1 DDX3 inhibition reverses HIV-1 latency in J-Lat 11.1 cells. a** J-Lat 11.1 cells were treated with increasing concentrations of the DDX3 inhibitor RK-33. % of reactivation, monitored by GFP production, was quantified by Flow cytometry. Error bars represent mean ± SD of three to nine independent experiments with at least 10,000 cells counted per treatment (paired, two-tailed *t*-test). The relative expression levels of (**b**) GFP mRNA and (**c**) vRNA in the RK-33-treated J-Lat cells were quantified by RT-qPCR. Error bars represent mean ± SD of three to five independent experiments (unpaired, two-tailed *t*-test; *p* value <0.00001). **d** Representative FISH-Flow dot plots from an independent experiment of J-Lat cells treated with DMSO (Mock), PMA or 2 μM RK-33. Gating strategy as depicted in Supplementary Fig. 1a. **e** The % of vRNA-expressing cells (GFP− or GFP+) as treated in **a** were quantified. Error bars represent mean ± SD of three independent experiments with at least 10,000 cells counted per treatment (unpaired, two-tailed *t*-test). Relative (**f**) MMP9 and (**g**) IL-8 mRNA expression levels from cells treated as in **a** were quantified by RT-qPCR. Error bars represent mean ± SD of three to four independent experiments (unpaired, two-tailed *t*-test).

pharmacological inhibition of DDX3 can affect proviral reactivation in a cell line model of HIV-1 latency, J-Lat 11.1. This model is a Jurkat-derived cell line that contains a latent integrated copy of the HIV-1 provirus defective in *env* with a GFP reporter in place of the *nef* open reading frame[39]. J-Lat 11.1 cells harboured a basal frequency of GFP positive cells at <7.5% that was reactivated by treatment with the PKC agonist phorbol myristate acetate (PMA) to >80% (Supplementary Fig. 1a). J-Lat 11.1 cells were treated with increasing concentrations (0.5–2 μM) of the DDX3 inhibitor RK-33 that has been developed for cancer treatment. RK-33 binds to the ATPase domain of DDX3 and is reported to improve radiation sensitisation of tumours in models of lung cancer, colorectal cancer, prostate cancer and Ewing's

carcinoma[27,28,40–44]. Interestingly, treatment with RK-33 alone induced viral reactivation in a dose-dependent manner, with ~18% of cells producing GFP when treated with 2 μM RK-33 (Fig. 1a), thus unravelling a role for DDX3 inhibitors as LRAs. A larger latency reversal activity of RK-33 was observed at the RNA level with a 70-fold and 25-fold increase in the relative expression of the HIV-1-promoter-controlled GFP and vRNA transcripts, respectively, as quantified by RT-qPCR (Fig. 1b, c). Since DDX3 has been reported to play important roles in the nucleocytoplasmic export and translation of HIV-1 vRNA[32,34–36], we hypothesised that RK-33 could also influence HIV-1 gene expression at a post-transcriptional level by impairing either vRNA export or translation. To investigate this,

we conducted FISH-Flow after treatment of J-Lat 11.1 cells with 2 μM RK-33 for 18 h. Using probes that amplify and allow identification and quantification of vRNA expression in cells, FISH-Flow allows for the simultaneous detection of distinct populations: (1) latent cells, which express neither viral RNA nor viral protein (vRNA−/GFP−), (2) efficiently reactivated cells, which express both HIV-1 viral RNA and proteins (vRNA+/GFP+), (3) transcriptionally competent viral RNA-expressing cells with either untranslated vRNA or a block in post-transcriptional vRNA processing and viral protein expression (vRNA+/GFP−) and (4) cells only expressing GFP from multiply spliced viral RNA transcripts (vRNA−/GFP+) (Fig. 1d). Thus, FISH-Flow gives an additional layer of insight on vRNA-regulation and given DDX3's previously characterised roles in post-transcriptional regulation of vRNA metabolism, FISH-Flow that uses RNA-expression as a readout is the most appropriate tool to investigate the effects of DDX3 inhibition on HIV-1. On average, under uninduced conditions, only 3% of cells constitutively produced vRNA, whereas vRNA was produced in more than 80% of cells upon PMA stimulation (Fig. 1d, e). Treatment with 2 μM RK-33 resulted in the expression of vRNA in ~15% of J-Lat cells (Fig. 1d, e), corroborating its latency reversal activity. Interestingly, ~8% of cells were observed in the vRNA+/GFP− quadrant, corresponding to ~42% of all vRNA+ cells (Fig. 1d, e). In contrast, merely 0.02% of the vRNA-expressing PMA-treated cells did not express GFP (Fig. 1d, e). These data indicate a post-transcriptional block in vRNA metabolism upon DDX3 inhibition. To determine if this block was at the level of nucleocytoplasmic export, we seeded the cells that underwent FISH-Flow analysis on a coverslip and observed them by confocal microscopy. In representative images, we observed that the vRNA+/GFP− cells in RK-33-treated conditions contained vRNA sequestered in the nucleus, while in double-positive cells in both the PMA and RK-33-treated condition, vRNA was present in the cytoplasm of cells (Supplementary Fig. 1b). GFP is expressed from the multiply spliced viral RNA transcript and the full-length vRNA codes for the viral protein pr55Gag[39]. Therefore, to further validate that RK-33 treatment affects post-transcriptional metabolism of the vRNA, we measured pr55Gag expression in cells treated with RK-33 by Western blot and did not observe significant upregulation of pr55Gag expression at a protein level (Supplementary Fig. 1c, d) despite vRNA expression at an RNA level (Fig. 1c–e). Therefore, DDX3 inhibition by RK-33 hinders nucleocytoplasmic export of the vRNA and viral protein expression, thereby highlighting the importance of using vRNA expression and not viral protein production as a marker for HIV-1 infection when investigating the effects of DDX3 inhibitors.

DDX3 has previously been reported to have cell-type-dependent differential effects on the regulation of the NF-κB pathways and has been implicated in binding to the p65 subunit thus inhibiting NF-κB-mediated transcription[45,46]. To determine if RK-33 is capable of activating NF-κB, Jurkat cells were nucleofected with firefly luciferase reporter plasmids either containing or lacking NF-κB consensus sites upstream of a minimal promoter, along with an internal Renilla control plasmid. Cells were then either mock-treated or treated with 2 μM RK-33 for 18 h. Treatment with RK-33 resulted in a modest increase in relative luciferase expression in the NF-κB reporter transfected cells as compared to the mock-treated cells (Supplementary Fig. 1e). The NF-κB target genes MMP9, IL-8 and IL-10 were also upregulated in J-Lat 11.1 cells treated with RK-33, suggesting activation of the NF-κB pathway (Fig. 1f, g and Supplementary Fig. 1f). Our data suggest that the DDX3 inhibitor RK-33 acts as a LRA that reactivates the HIV-1 provirus, possibly via activation of NF-κB.

**DDX3 inhibition results in increased cell death in vRNA-expressing cells.** In order to further characterise the impact of DDX3 inhibition and, hence DDX3 function, in HIV-1 latency, we performed shRNA-depletion of DDX3 in J-Lat 11.1 cells and monitored latency reversal and NF-κB target upregulation. Lentiviral transduction of J-Lat 11.1 cells with shRNA against DDX3 (shDDX3) resulted in a 90% decrease in DDX3 mRNA expression and an 80% decrease in DDX3 protein expression compared to the cells treated with scrambled shRNA (shControl) as quantified by RT-qPCR and Western blot, respectively (Fig. 2a, b and Supplementary Fig. 2a). DDX3-depleted cells demonstrated a twofold increase in GFP expression as compared to the shControl transduced cells (Fig. 2c), thereby confirming that inhibition of DDX3 reverses HIV-1 latency. Latency reversal (vRNA and GFP mRNA) and NF-κB target upregulation (IL-10, IL-8 and MMP9) were also measured by RT-qPCR in DDX3-depleted cells and a significant increase in GFP mRNA and vRNA expression was observed compared to mock-treated cells, reaffirming latency reversal at a transcriptional level (Fig. 2d). Moreover, similar to our observations with the chemical inhibition of DDX3 with RK-33, NF-κB targets genes IL-10, IL-8 and MMP9 were significantly upregulated in DDX3-depleted cells as compared to the control (Fig. 2d).

During viral infection, single-stranded RNA is recognised by RIG-I that induces downstream innate antiviral responses mediated by NF-κB and IRF3 resulting in IFN-β expression[47–49]. We investigated whether these effectors are activated by HIV-1 vRNA in the absence of DDX3. shRNA-mediated depletion of DDX3 resulted in a significant twofold increase in IFN-β mRNA expression compared to the mock-treated cells (Fig. 2d). Similarly, chemical inhibition of DDX3 using 2 μM RK-33 resulted in modest upregulation of IFN-β expression in J-Lat 11.1 cells (Supplementary Fig. 2b). Interestingly, Jurkat cells that do not express vRNA did not demonstrate any significant upregulation of IFN-β upon RK-33 treatment, suggesting that IFN-β induction is dependent on vRNA expression (Supplementary Fig. 2b).

To enrich for vRNA-expressing cells and examine the specific contribution of HIV-1 vRNA to the activation of innate immune pathway components during DDX3 inhibition, we sorted J-Lat 11.1 cells into vRNA low and vRNA high populations by FACS following treatment with 2 μM RK-33 as shown (Fig. 2e). Our gating strategy was based on GFP expression (Supplementary Fig. 1a), which resulted in two populations of cells; vRNA low (in which 96% of cells were vRNA negative) and vRNA high (where 68% of cells were vRNA positive). As expected, the vRNA high J-Lat 11.1 sorted cells contain ~300-fold more vRNA than the vRNA low population (Fig. 2f). The vRNA high population demonstrated induction of the innate antiviral signalling pathway components RIG-I and MAVS, consistent with previous work demonstrating HIV-1 vRNA's ability to activate these pathways (Fig. 2g)[17,18]. Interestingly, concomitant significant phosphorylation of IRF3 was also observed in the vRNA-high expressing cells, confirming the activation of innate antiviral signalling pathways in vRNA-expressing cells (Fig. 2h, i). The phosphorylation of IRF3 in combination with NF-κB activation results in the production of type I interferons[47–49]. In agreement, we observed a significant increase in IFN-β mRNA expression in the vRNA-high cell population (Fig. 2j). The characterised role of IFN-β in contributing to apoptosis, also during viral infection[50–52] together with RK-33's described roles in inducing cell death[40], prompted us to examine whether the vRNA high cells are more susceptible to apoptosis. Concomitant with the observed increased viral RNA expression, we also observed an ~35% decrease in the ratio of the anti-apoptotic BCl2 to the pro-apoptotic Bax mRNA expression levels in the HIV-1 vRNA-high population, indicating that the

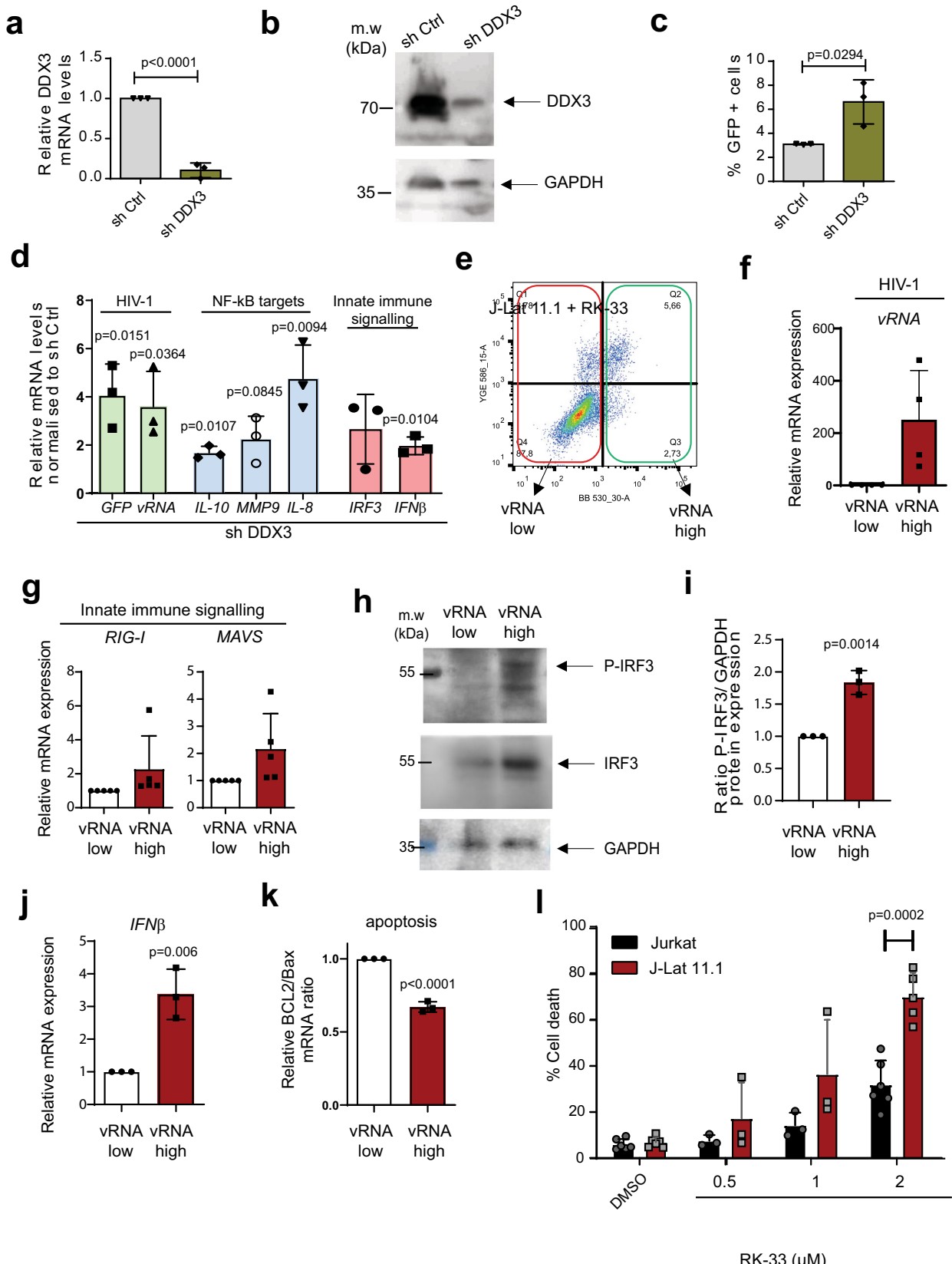

vRNA-expressing cells have a higher susceptibility to undergo apoptosis (Fig. 2k). To further probe the selectivity of cell death, J-Lat 11.1 cells and parental Jurkat cells lacking integrated HIV-1 provirus and vRNA were treated with increasing concentrations of RK-33 (0.5–2 µM). Eighteen hours post-treatment, cells were

stained with Hoescht 33342 and the percentage of viable cells was quantified by flow cytometry. We observed a dose-dependent increase in cell death in RK-33-treated J-Lat cells compared to parental Jurkat cells, most strikingly at 2 µM RK-33 (Fig. 2i). In this condition, cell death was observed in 70% of J-Lat 11.1 cells,

**Fig. 2 DDX3 inhibition induces IRF3 phosphorylation, IFNβ expression and cell death in J-Lat 11.1 cells. a** J-Lat 11.1 cells were transduced with a control lentiviral vector expressing a scrambled shRNA (shControl) or with lentivirus expressing shRNA against DDX3 (shDDX3) and relative DDX3 mRNA expression was quantified by RT-qPCR. Error bars represent the mean ± SD of three independent experiments (unpaired, two-tailed *t*-test; *p* value = 0.001038). **b** Representative Western blots depicting DDX3 and GAPDH in shControl and shDDX3 conditions. **c** The % of reactivation monitored as GFP production from cells as treated in **a** were quantified by Flow cytometry. Gating strategy as depicted in Supplementary Fig. 1a. Error bars represent mean ± SD of three independent experiments with at least 10,000 cells counted per treatment (paired, two-tailed *t*-test). **d** In cells treated as in **a**, relative GFP, vRNA, IL-10, IL-8, MMP9, IRF3 and IFNβ mRNA expression levels in shDDX3 treated cells were quantified by RT-qPCR and normalised to the shControl. Error bars represent mean ± SD of three independent experiments (unpaired, two-tailed *t*-test). **e** RK-33-treated J-Lat cells were sorted into GFP− (vRNA low) and GFP+ (vRNA high) fractions and the relative mRNA expression of vRNA using primers against (**f**) the vRNA and (**g**) RIG-I and MAVS expression were quantified by RT-qPCR. **h** Representative Western blots depicting phosphorylated IRF3, total IRF3 and GAPDH protein expression in RK-33-treated vRNA low and vRNA high populations. **i** Quantification of the densitometric analysis of P-IRF3 to GAPDH protein expression ratios from 2 h. Error bars represent mean ± SD from three independent experiments (unpaired, two-tailed *t*-test). The relative expression of (**j**) IFNβ mRNA and (**k**) ratios of Bcl2 and Bax mRNA expression in RK-33-treated vRNA low and vRNA high populations were quantified by RT-qPCR. Error bars represent mean ± SD from three independent experiments (unpaired, two-tailed *t*-test). **l** % Cell death was measured in Jurkat vs. J-Lat cells treated with increasing concentrations of RK-33 by flow cytometry. Error bars represent the mean ± SD of at least three independent experiments with at least 10,000 cells counted per treatment (paired, two-tailed *t*-test).

---

but only in 30% of Jurkat cells representing a significant increase in cell death in HIV-1-infected cells. Since they are a cancer-derived cell line, the viability of Jurkat cells was also affected upon treatment with the RK-33, albeit not as much as the J-Lat 11.1 cells (Supplementary Fig. 2c). In comparison, RK-33 had negligible effects on the viability of primary CD4+ T cells (Supplementary Fig. 2c). Our data, therefore, indicate that DDX3 inhibition results in latency reversal, upregulation of NF-κB target genes and IRF3, leading to induction of IFN-β expression and a pro-apoptotic state, thereby rendering vRNA-expressing cells more susceptible to cell death during vRNA expression.

**DDX3 inhibition downregulates genes involved in cell survival.** The inhibition of DDX3 helicase function, i.e., dsRNA unwinding, can be achieved by competitive inhibition of the ATP or RNA-binding sites. RK-33 was demonstrated to inhibit RNA helicase activity in a concentration-dependent manner reducing ATP consumption in enzymatic assays[40]. A class of inhibitors competitive for the DDX3 RNA-binding site have been reported by Radi et al.[53] and were further developed leading to the identification of the potent and selective compound FH-1321 (compound 16d in[38]), that specifically targets the DDX3 RNA-binding site with no effect in its ATPase activity[38]. A hypothetical binding model obtained by molecular docking calculation for both RK-33 and FH-1321 depicts the predicted binding of RK-33 to the ATP-binding site of DDX3, while FH-1321 binds the RNA-binding site (Fig. 3a and Supplementary Fig. 3a, b). For both molecules, the downstream effect is inhibition of the RNA helicase activity of DDX3[40,53]. We, therefore, investigated the potential function and mechanism of action of RK-33 and FH-1321 in HIV-1 reactivation and reservoir dynamics in the more relevant primary CD4+ T cells. We observed no significant decrease in viability when primary CD4+ T cells from three healthy donors were treated with up to 10 μM of either compound (Supplementary Fig. 3c). We also cultured CD4+ T cells in the presence or absence of 2 μM RK-33 or 1 μM FH-1321, for 5 days and observed no significant differences in cell viability between the DDX3 inhibitor-treated and mock-treated conditions (Supplementary Fig. 3d). To gain insight into the molecular mechanism by which DDX3 inhibition may ICD in HIV-1-infected cells, we treated primary CD4+ T cells obtained from two healthy donors with RK-33 and FH-1321 and performed RNA-sequencing. Minimal changes in gene expression were observed 4 h post-treatment with 2 μM RK-33 or 1 μM FH-1321 with one and five differentially expressed genes, respectively, with a 1.5-fold difference in expression, and a false discovery cut-off rate set to 0.05 (Supplementary Fig. 3e, f). At 18 h post-treatment: primary CD4+ T cells treated with 1 μM

FH-1321 displayed only 34 genes differentially expressed, while treatment with RK-33 (2 μM) resulted in 39 genes differentially expressed in both donors (Fig. 3b–d and Tables 1 and 2). Interestingly, the majority of the differentially expressed genes were common to both the RK-33 and FH-1321 treatments (*n* = 23). Since these effects are common between compounds belonging to two different classes of DDX3 inhibitors, they are bound to be specific to the inhibition of DDX3 helicase activity (Fig. 3e). Notably, the overall change in gene expression was limited, consistent with the minimal potential for off-target or cytotoxic effects associated with the use of these compounds. Interestingly, two of the differentially expressed genes significantly down-regulated in CD4+ T cells in response to RK-33, the inhibitor of apoptosis protein BIRC5 and HSPA1B (Hsp70), have been previously implicated to play a role in HIV-1 induced cell death. BIRC5, which is also significantly downregulated in response to FH-1321 treatment, has been previously demonstrated to be upregulated in latent HIV-1-infected cells, and its inhibition resulted in increased cell death of HIV-1-infected cells[54]. Also involved in regulating apoptosis in infected cells, Hsp70 was shown to inhibit apoptosis induced by the HIV-1 protein Vpr[55]. Consistent with the observed DDX3-mediated upregulation of NF-κB target genes, HSP70 was also shown to inhibit HIV-1 gene expression[56], possibly via suppression of NF-κB signalling through its interaction with HSP70 binding protein 1[57]. RK-33 and FH-1321-mediated downregulation of BIRC5 and RK-33-mediated HSPA1B downregulation were confirmed at the RNA level by RT-qPCR in six additional uninfected donor CD4+ T cells (Fig. 3f) and at the protein level by Western blot analysis in CD4+ T cells from three donors (Fig. 3g–i and Supplementary Fig. 3g). We also conducted shRNA-mediated depletion of DDX3 in CD4+ T cells from three donors and observed a decrease in DDX3 protein expression upon shDDX3-containing lentiviral transduction (Fig. 3j and Supplementary Fig. 3h). This also corresponded to a decrease in expression of BIRC5 at both protein and mRNA levels, but no decrease in HSPA1B at an mRNA level (Fig. 3j and Supplementary Fig. 3h, i). To ensure that DDX3 inhibitors do not induce T cell activation, we examined CD25 expression upon treatment with RK-33 and FH-1321 (Supplementary Fig. 3j). Consistent with the minimal effects observed with the RNA-sequencing, treatment with DDX3 inhibitors did not result in the induction of T cell activation, nor did DDX3 inhibition compromise effector function in CD4+ T cells (Supplementary Fig. 3k, l) or CD8+ T cells (Supplementary Fig. 4a, b) as determined by flow cytometry measuring IL-2 and IFNγ upon PMA-induced activation. Moreover, treatment with DDX3 inhibitors did not affect the capacity for proliferation in either

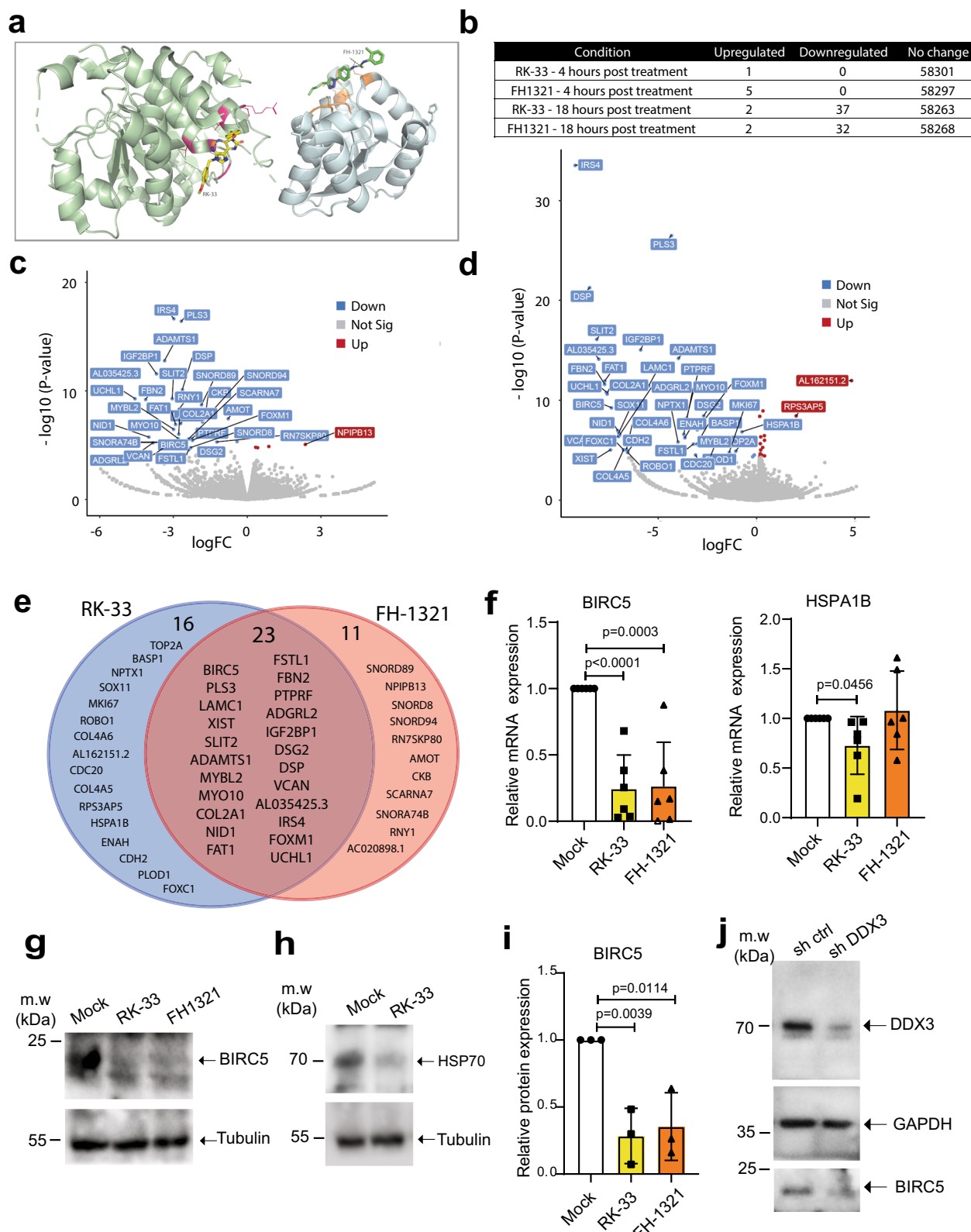

CD4+ T or CD8+ T cells (Supplementary Fig. 4c–f) as determined after induction/stimulation with CD3/CD28. We also confirmed that DDX3 inhibition at the concentrations used does not affect de novo host cell translation in primary CD4+ T cells (Supplementary Fig. 4g, h). Finally, we determined if treatment of CD4+ T cells with DDX3 inhibitors could induce an upregulation of NF-κB target genes by RT-qPCR. Treatment with RK-33

and FH-1321 resulted in induction of the targets IL-8 and MMP9 mRNA expression, and whereas FH-1321 treatment induced expression of IL-8 and IL-10 mRNA expression (Supplementary Fig. 4i–k).

**DDX3 inhibitors reactivate and selectively kill HIV-1-infected cells in a primary model of latency.** Given our observations that

**Fig. 3 Effects of DDX3 inhibition on CD4+ T cells. a** Molecular docking of RK-33 and FH-1321 to pre-unwound DDX3 confirmation as predicted by AutoDock Vina. DDX3 domain 1 is shown as a pale green cartoon, domain 2 is shown as a pale blue cartoon and the surface colour code is pink for ATP-binding residues and orange for RNA-binding residues. The colour code for RK-33 is yellow for carbon, red for oxygen, dark blue for nitrogen. The colour code for FH-1321 is green for carbon, red for oxygen, dark blue for nitrogen. The predicted hydrogen bond between FH-1321 and Lysine 541 from the DDX3 RNA-binding pocket is indicated with a dashed line. **b** Table of differentially expressed genes upon treatment of CD4+ T cells with DDX3 inhibitors. Volcano plot of differentially expressed genes 18 h post-treatment with (**c**) RK-33 and (**d**) FH-1321. **e** Venn diagram of overlapping differentially expressed genes 18 h post-treatment with DDX3 inhibitors. **f** CD4+ T cells from healthy donors were treated with DMSO (Mock), RK-33 and FH-1321 and the relative expression of BIRC5 and HSPA1B mRNAs were quantified by RT-qPCR. Error bars represent the mean ± SD of independent experiments with cells from six different donors (unpaired, two-tailed t-test; p value for BIRC5 expression in RK-33 treated cells is 0.000027). **g** Representative Western blots of CD4+ T cells treated with 2 μM RK-33 or 1 μM FH-1321 depicting BIRC5 and (**h**) HSP70 expression with loading controls as indicated. **i** Quantification of the densitometric analysis of BIRC5 protein expression from cells as treated in Fig. 2g. Error bars represent the mean ± SD from three donors (unpaired, two-tailed t-test). **j** Representative Western blots depicting DDX3, BIRC5 and GAPDH protein levels from CD4+ T cells stimulated with CD3-CD28-coated beads and infected with a control lentiviral vector (shControl) or a lentiviral vector expressing shRNA against DDX3 (shDDX3).

| Table 1 EdgeR report of differentially expressed genes 18 h post-treatment with 2 μM RK-33. | | | | | | |
|---|---|---|---|---|---|---|
| GeneID | Name | log2 fold change | log2 counts per million | *F* value | *p* value | False discovery rate |
| ENSG00000133124 | IRS4 | −9.220240267 | 1.229125796 | 149.1749 | 2.89E−34 | 1.69E−29 |
| ENSG00000096696 | DSP | −8.484792295 | 0.549901848 | 93.17747 | 4.97E−22 | 9.65E−18 |
| ENSG00000145147 | SLIT2 | −8.122797551 | 0.215628794 | 69.49186 | 7.84E−17 | 1.14E−12 |
| ENSG00000261409 | AL035425.3 | −8.022265953 | 0.113605342 | 60.57704 | 7.19E−15 | 5.99E−11 |
| ENSG00000138829 | FBN2 | −7.721874417 | −0.141878891 | 49.16692 | 2.38E−12 | 1.39E−08 |
| ENSG00000083857 | FAT1 | −7.711916601 | −0.142206665 | 50.58668 | 1.15E−12 | 7.47E−09 |
| ENSG00000154277 | UCHL1 | −7.629659662 | −0.220071849 | 45.5358 | 1.51E−11 | 7.35E−08 |
| ENSG00000139219 | COL2A1 | −7.594048405 | −0.247211954 | 44.80238 | 2.20E−11 | 9.86E−08 |
| ENSG00000229807 | XIST | −7.425086724 | 4.779411144 | 281.6285 | 8.79E−06 | 0.013432 |
| ENSG00000089685 | BIRC5 | −7.372112695 | −0.421255852 | 38.53409 | 5.42E−10 | 2.26E−06 |
| ENSG00000116962 | NID1 | −7.05272162 | −0.69051715 | 28.24878 | 1.07E−07 | 0.000297 |
| ENSG00000038427 | VCAN | −7.017959819 | −0.727725687 | 25.70546 | 3.99E−07 | 0.000861 |
| ENSG00000176887 | SOX11 | −7.004754934 | −0.728056321 | 27.13324 | 1.91E−07 | 0.000483 |
| ENSG00000117114 | ADGRL2 | −6.95027203 | −0.766705238 | 26.38183 | 2.81E−07 | 0.000655 |
| ENSG00000054598 | FOXC1 | −6.756514098 | −0.931471077 | 19.71939 | 8.99E−06 | 0.013432 |
| ENSG00000188153 | COL4A5 | −6.756514098 | −0.931471077 | 19.71939 | 8.99E−06 | 0.013432 |
| ENSG00000197565 | COL4A6 | −6.610183248 | −1.02235097 | 19.90055 | 8.17E−06 | 0.013236 |
| ENSG00000169855 | ROBO1 | −6.551036428 | −1.069611081 | 18.62718 | 1.59E−05 | 0.021095 |
| ENSG00000170558 | CDH2 | −6.544969004 | −1.069721671 | 18.90017 | 1.38E−05 | 0.019153 |
| ENSG00000159217 | IGF2BP1 | −5.889308666 | 0.348421941 | 64.80318 | 8.72E−16 | 1.02E−11 |
| ENSG00000135862 | LAMC1 | −5.486512767 | 0.002490741 | 48.29217 | 3.71E−12 | 1.97E−08 |
| ENSG00000102024 | PLS3 | −4.323906265 | 1.49827749 | 117.2158 | 3.02E−27 | 8.81E−23 |
| ENSG00000171246 | NPTX1 | −3.992292706 | −0.550303193 | 23.25519 | 1.42E−06 | 0.002675 |
| ENSG00000154734 | ADAMTS1 | −3.971030294 | 0.580596315 | 60.82655 | 6.34E−15 | 5.99E−11 |
| ENSG00000145555 | MYO10 | −3.696600515 | −0.303481708 | 26.65852 | 2.44E−07 | 0.000592 |
| ENSG00000154380 | ENAH | −3.64365296 | −0.274861002 | 25.06365 | 5.59E−07 | 0.001124 |
| ENSG00000142949 | PTPRF | −3.553235332 | −0.021528142 | 33.79895 | 6.35E−09 | 1.95E−05 |
| ENSG00000163430 | FSTL1 | −3.20753747 | −0.332407149 | 22.31002 | 2.33E−06 | 0.003987 |
| ENSG00000117399 | CDC20 | −3.098292124 | −0.420984699 | 17.47004 | 2.92E−05 | 0.034778 |
| ENSG00000176788 | BASP1 | −3.03195077 | −0.24810378 | 22.38461 | 2.24E−06 | 0.003952 |
| ENSG00000111206 | FOXM1 | −3.015861999 | 0.001271235 | 28.7242 | 8.38E−08 | 0.000244 |
| ENSG00000046604 | DSG2 | −2.684610226 | 0.484772447 | 34.90713 | 3.48E−09 | 1.19E−05 |
| ENSG00000101057 | MYBL2 | −2.316037106 | 0.025267442 | 17.38763 | 3.06E−05 | 0.035688 |
| ENSG00000148773 | MKI67 | −1.384356143 | 1.091562301 | 18.66531 | 1.56E−05 | 0.021095 |
| ENSG00000083444 | PLOD1 | −1.18441556 | 1.399169378 | 17.50631 | 2.88E−05 | 0.034778 |
| ENSG00000131747 | TOP2A | −1.054906153 | 1.866051914 | 19.13111 | 1.25E−05 | 0.017772 |
| ENSG00000204388 | HSPA1B | −0.741213605 | 3.325135537 | 27.77549 | 1.39E−07 | 0.000367 |
| ENSG00000178429 | RPS3AP5 | 2.015230135 | 1.279297836 | 36.33718 | 3.41E−09 | 1.19E−05 |
| ENSG00000234648 | AL162151.2 | 4.827847593 | 0.344327701 | 50.73859 | 1.07E−12 | 7.47E−09 |

Differential expression and p values are calculated for each gene using an exact test analogous to Fisher's exact test adapted for overdispersed data in the EdgeR package[88].

DDX3 inhibition reverses latency, induces more cell death in the J-Lat model and does not affect primary CD4+ T cell function, we wondered if we could use the DDX3 inhibitors to reactivate the provirus and ICD in a primary CD4+ T cell model of HIV-1 latency, developed by Lassen et al.[58–60] (Fig. 4a). In this model of HIV-1 latency, the virus used to infect the primary CD4+ T cells contained a luciferase reporter inserted in the *nef* open reading frame, which is encoded by the multiply spliced vRNA transcript (similar to the GFP in the J-Lat cells). We treated latently infected cells with 2 μM RK-33, 1 μM FH-1321 or PMA-ionomycin (as a positive control) and monitored latency reversal 18 h post-treatment by FISH-Flow[22], RT-qPCR and relative luciferase expression (Fig. 4a). To quantify ICD, we also co-stained cells with a fixable viability dye and either with a fluorophore-conjugated

**Table 2 EdgeR report of differentially expressed genes 18 h post-treatment with 1 μM FH-1321.**

| GeneID | Name | log2 fold change | log2 counts per million | F value | p value | False discovery rate |
|---|---|---|---|---|---|---|
| ENSG00000154277 | UCHL1 | −4.565471687 | −0.206793672 | 38.67266739 | 5.05E−10 | 3.31E−06 |
| ENSG00000261409 | AL035425.3 | −4.424650789 | 0.138068277 | 48.33716477 | 3.63E−12 | 3.52E−08 |
| ENSG00000138829 | FBN2 | −4.131894042 | −0.10684436 | 38.38083941 | 5.86E−10 | 3.42E−06 |
| ENSG00000116962 | NID1 | −3.999181717 | −0.658930046 | 22.77369298 | 1.83E−06 | 0.0053254 |
| ENSG00000117114 | ADGRL2 | −3.832053102 | −0.731611642 | 20.4765316 | 6.05E−06 | 0.0125945 |
| ENSG00000159217 | IGF2BP1 | −3.68912559 | 0.398606752 | 48.96044087 | 2.64E−12 | 3.08E−08 |
| ENSG00000229807 | XIST | −3.488985213 | 4.901654584 | 651.1533715 | 7.66E−143 | 4.47E−138 |
| ENSG00000038427 | VCAN | −3.429296401 | −0.658534387 | 18.19677341 | 2.00E−05 | 0.0323138 |
| ENSG00000154734 | ADAMTS1 | −3.363272907 | 0.603352768 | 54.52344803 | 1.56E−13 | 2.27E−09 |
| ENSG00000212402 | SNORA74B | −3.359506375 | 0.031288549 | 24.79248338 | 5.34E−06 | 0.0115336 |
| ENSG00000145147 | SLIT2 | −3.051459541 | 0.34517457 | 38.64546164 | 5.12E−10 | 3.31E−06 |
| ENSG00000101057 | MYBL2 | −2.996889004 | −0.083099003 | 26.07056348 | 3.34E−07 | 0.0010805 |
| ENSG00000133124 | IRS4 | −2.98565677 | 1.371029055 | 71.98000489 | 2.22E−17 | 6.48E−13 |
| ENSG00000083857 | FAT1 | −2.913815462 | −0.014539277 | 28.05815995 | 1.18E−07 | 0.0004052 |
| ENSG00000145555 | MYO10 | −2.805586002 | −0.261404374 | 22.18212809 | 2.49E−06 | 0.0066275 |
| ENSG00000139219 | COL2A1 | −2.783583292 | −0.107869982 | 24.02115361 | 9.55E−07 | 0.0029316 |
| ENSG00000201098 | RNY1 | −2.770572459 | 0.413772602 | 35.10682622 | 4.80E−09 | 2.33E−05 |
| ENSG00000135862 | LAMC1 | −2.712320987 | 0.135877854 | 28.32916412 | 1.03E−07 | 0.0003743 |
| ENSG00000102024 | PLS3 | −2.681318527 | 1.605768497 | 71.57130553 | 3.77E−17 | 7.32E−13 |
| ENSG00000096696 | DSP | −2.636756013 | 0.730407853 | 42.41260665 | 7.45E−11 | 6.21E−07 |
| ENSG00000089685 | BIRC5 | −2.490957592 | −0.234929297 | 18.96212263 | 1.34E−05 | 0.02512 |
| ENSG00000238741 | SCARNA7 | −2.387719434 | 0.361869871 | 24.20859794 | 2.61E−06 | 0.0066275 |
| ENSG00000163430 | FSTL1 | −2.380561336 | −0.26218126 | 17.71426598 | 2.57E−05 | 0.0405086 |
| ENSG00000200785 | SNORD8 | −2.299517907 | −0.084333496 | 18.58796384 | 1.64E−05 | 0.0290541 |
| ENSG00000142949 | PTPRF | −2.256619192 | 0.113519171 | 22.13114508 | 2.55E−06 | 0.0066275 |
| ENSG00000111206 | FOXM1 | −2.226389714 | 0.092218251 | 21.57249486 | 3.41E−06 | 0.0082946 |
| ENSG00000212283 | SNORD89 | −1.860714666 | 1.311190667 | 36.42603494 | 1.90E−09 | 1.01E−05 |
| ENSG00000208772 | SNORD94 | −1.822933237 | 1.022233292 | 29.05095049 | 7.20E−08 | 0.0002797 |
| ENSG00000166165 | CKB | −1.686278111 | 1.500619457 | 33.19463106 | 9.30E−09 | 4.17E−05 |
| ENSG00000046604 | DSG2 | −1.609604699 | 0.644434181 | 18.28993873 | 1.90E−05 | 0.031651 |
| ENSG00000202058 | RN7SKP80 | −1.234932643 | 1.641387087 | 21.21111392 | 4.96E−06 | 0.011119 |
| ENSG00000126016 | AMOT | −0.773124251 | 3.662624092 | 30.45093151 | 3.53E−08 | 0.0001469 |
| ENSG00000130164 | LDLR | −0.411325614 | 5.764550633 | 21.07423587 | 4.43E−06 | 0.0103253 |
| ENSG00000101745 | ANKRD12 | 0.339396244 | 7.502647544 | 18.65815618 | 1.57E−05 | 0.0285379 |
| ENSG00000170881 | RNF139 | 0.423776234 | 5.074578288 | 18.46989874 | 1.74E−05 | 0.0297963 |
| ENSG00000213757 | AC020898.1 | 0.876931574 | 2.474353677 | 19.06819338 | 1.27E−05 | 0.0246334 |
| ENSG00000198064 | NPIPB13 | 2.365643737 | 0.043629688 | 19.87731825 | 8.31E−06 | 0.0167005 |

Differential expression and p values are calculated for each gene using an exact test analogous to Fisher's exact test adapted for overdispersed data in the EdgeR package[88].

antibody that detects AnnexinV or cleaved Caspase-3/7, both markers for host cell apoptosis (Gating strategy in Supplementary Fig. 5a). FISH-Flow has a high specificity for the target HIV-1 vRNA with a negligible background in the uninfected CD4+ T cells and a sensitivity to detect up to 10 HIV-1 vRNA+ cells per million cells[16,22] (Fig. 4b). In latent HIV-1-infected CD4+ T cells, basal levels of vRNA were produced in the uninduced (Mock) condition (~165 vRNA+ cells/million), with a frequency of >650 vRNA+ cells/million upon PMA-ionomycin stimulation, corresponding to a 4.8 (s.d. ± 3.3) fold increase (Fig. 4b, c and Supplementary Fig. 5b). Treatment with the DDX3 inhibitors RK-33 and FH-1321 resulted in ~410 vRNA+ cells/million and ~365 vRNA+ cells/million, respectively, corresponding to a 2.8 (s.d. ± 1.1) fold and a 2.5 (s.d. ± 1.4) fold increase in the frequency of vRNA-expressing cells as compared to the mock-treated condition (Fig. 4b, c and Supplementary Fig. 5b). We also measured intracellular vRNA expression and HIV-1 multiply spliced RNA (msRNA) from cell lysates by RT-qPCR and observed a 12.4-fold increase in vRNA expression with RK-33, a 15.7-fold increase with FH-1321 and a 66.8-fold increase with PMA-ionomycin as compared to the mock-treated cells (Fig. 4d and Supplementary Fig. 5c). When treated with DDX3 inhibitors, a 1.5-fold increase in relative luciferase units as compared to the mock-treated condition was observed in both RK-33 and FH-1321-treated conditions (Fig. 4e). DDX3 influences viral gene expression at a post-transcriptional level (Fig. 1e). In agreement, the observed fold increase in luciferase expression upon DDX3 inhibition was not as high as that of vRNA expression. Together, these data demonstrate that the DDX3 inhibitors also induce latency reversal in a primary CD4+ T cell model of HIV-1 latency. Based on our observations of increased cell death in J-Lat 11.1. cells in comparison to the Jurkat cells minimal cell death at the working concentrations in uninfected primary CD4 T cells (Fig. 2i and Supplementary Fig. 3c, d), we sought to determine if increased cell apoptosis could be observed specifically in the vRNA-expressing primary CD4+ T cells in the in vitro model of HIV-1 latency. For this, we quantified the percentage of AnnexinV positive cells in cultures treated with mock, RK-33, FH-1321 or PMA-Ionomycin. Because FISH-Flow is a single-cell technique, we could distinctly quantify the differential expression of the vRNA+ cells or the vRNA−/bystander cells from our latency model. To validate the selectivity of cell death upon drug treatment, we also quantified the percentage of AnnexinV expression in uninfected CD4+ T cells. Basal levels of AnnexinV and Caspase-3 positivity (<1%) were observed in the uninfected cells or the vRNA− bystander cells (Fig. 4f, g). Of the vRNA+ cells, 4.8 (s.d. ± 4)% were AnnexinV positive in the cultures treated with PMA-ionomycin, indicating that viral reactivation might already make the cells more susceptible to cell death (Fig. 4f). Interestingly, in the cultures treated with

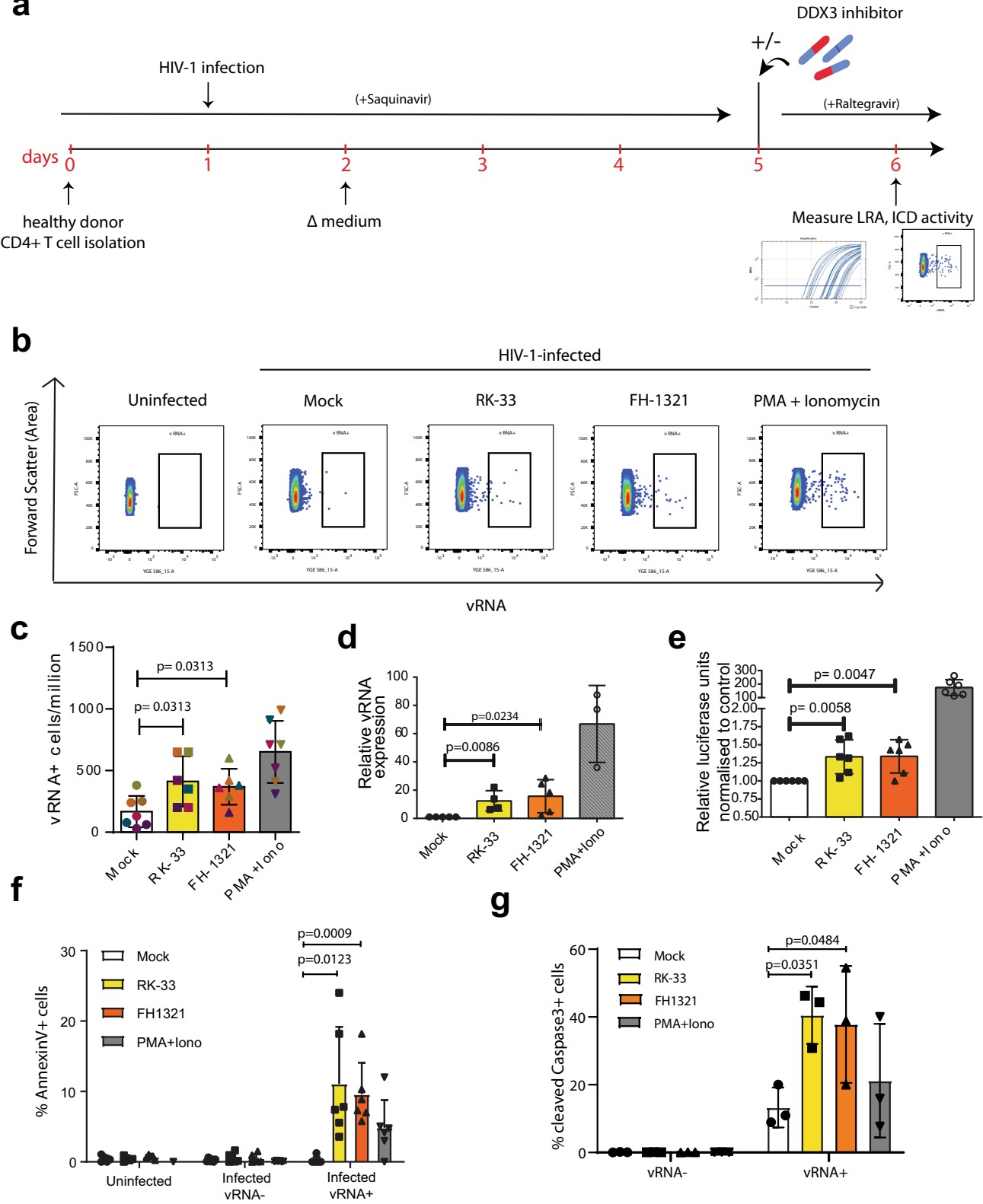

RK-33 and FH-1321, 11.1 (s.d. ± 8.1)% and 9.6 (s.d. ± 4.5)% of vRNA-expressing cells respectively, were AnnexinV positive (Fig. 4f). To further validate that the vRNA+ cells are indeed undergoing apoptosis, we repeated these studies in three additional donors and measured the expression of cleaved Caspase-3/7 by FISH-Flow. We observed that 40.5 (s.d. ± 8.5)% and 38.9 (s.d. ± 15.5)% of RK-33 and FH-1321 treated cells were cleaved Caspase-3/

7 positive as compared to the 13.3 (s.d. ± 5.9)% positive cells in the Mock-treated condition (Fig. 4g). To determine if BIRC5 down-regulation plays a role in latency reversal, we treated J-Lat 11.1 cells with the BIRC5 inhibitor YM155[61], but did not observe significant latency reversal, suggesting that BIRC5 plays a more prominent role in the death of vRNA-expressing cells rather than their latency reversal (Supplementary Fig. 5d). We were unable to observe vRNA

**Fig. 4 DDX3 inhibition has LRA and ICD activity in a primary model of HIV-1 latency. a** Schematic for latency establishment in primary human CD4+ T cells using the Lassen and Greene model. Latent HIV-1-infected cells were treated on day 5 with DMSO (unstimulated), PMA/Ionomycin, and the DDX3 inhibitors 2 μM RK-33 and 1 μM FH-1321. Uninfected cells were used as negative control. **b** Representative FISH-Flow dot plots of cells as treated in **a** representing 50,000 cells from each condition. Gating strategy depicted in Supplementary Fig. 5a. **c** The frequency of vRNA-expressing cells per million as depicted in **b** were quantified. Individual donors are represented by the same colour and symbols represent treatments (Mock = circles, RK-33 = squares, FH-1321 = upward triangles and PMA = downward triangles). Error bars represent the mean ± SD of independent experiments using cells from six to seven different donors, with a minimum of 100,000 events acquired for each condition (paired, two-tailed, Wilcoxon test). **d** The relative vRNA expression levels from cells as treated in **a** were quantified by RT-qPCR. Error bars represent the mean ± SD of independent experiments with cells from three to five different donors (unpaired, two-tailed *t*-test). **e** Relative luciferase activity of cells as treated in **a** was measured and normalised to the mock-treated condition. Error bars represent the mean ± SD of independent experiments with cells from six different donors (unpaired, two-tailed *t*-test). **f** The percentages of AnnexinV positive cells from the uninfected, infected vRNA− fractions, or infected vRNA+ cells as quantified by FISH-Flow. Error bars represent the mean ± SD of independent experiments with cells from six different donors, with a minimum of 100,000 events acquired for each condition (paired, two-tailed *t*-test). **g** The percentages of cleaved Caspase-3 positive cells from the infected vRNA− fractions or infected vRNA+ cells as quantified by FISH-Flow. Error bars represent the mean ± SD of independent experiments with cells from three different donors, with a minimum of 100,000 events acquired for each condition (paired, two-tailed *t*-test).

+ mediated effects on the relative expression of the anti-apoptotic protein Bcl2 and the pro-apoptotic protein Bax by RT-qPCR (Supplementary Fig. 5e) since only a small fraction of cells (<1%) is HIV-1-infected within the total population, further highlighting the importance of FISH-Flow which enables investigation of HIV-1-induced selective cell death at the single-cell level. Together, these data show that DDX3 inhibitors exacerbate apoptosis triggered by the vRNA and selectively ICD in vRNA-expressing cells, while uninfected cells are not affected.

**DDX3 inhibitors induce latency reversal in CD4+ T cells from HIV-1-infected donors**. To further investigate the clinical potential of DDX3 inhibitors we examined their ability to induce latency reversal and selective cell death in CD4+ T cells from chronic HIV-1-infected donors on suppressive cART with undetectable viral loads for at least 12 months (minimum $n = 6$). Following isolation from cryopreserved PBMCs obtained by leukapheresis of HIV-1-infected donors, CD4+ T cells were treated according to the schematic in Fig. 5a. Briefly, cells were cultured either with mock treatment, 2 μM RK-33 or 1 μM FH-1321 overnight in the presence of an HIV-1 integrase inhibitor (Raltegravir). Cells were collected 18 h post-treatment and analysed by FISH-Flow using probes against the vRNA, a viability stain and antibodies directed against AnnexinV to quantify viral reactivation and selective early apoptosis using a similar gating strategy as described in Supplementary Fig. 5a. A nested RT-qPCR was also performed to measure cell-associated vRNA with primers targeting the *Gag* region of the vRNA. Compared to a frequency of 9.5 (s.d. ± 4.8) vRNA-expressing cells per million (mock-treated conditions), 28.9 (s.d. ± 9), 22 (s.d. ± 4.5) and 52.8 (s.d. ± 22.6) vRNA+ cells per million were observed in the RK-33, FH-1321 and PMA-Ionomycin treated cells, respectively; corresponding to a 3.6 (s.d. ± 12.1) fold and a 2.7 (s.d. ± 1) fold increase in the percentage of vRNA-producing cells in RK-33 and FH-1321-treated conditions respectively (Fig. 5b, c and Supplementary Fig. 6a). Importantly, 28.6 (s.d. ± 13.5)% and 34.8 (s.d. ± 10.3)% of vRNA+ cells in the RK-33 and FH-1321 conditions were AnnexinV positive, while all the vRNA- populations had <1% AnnexinV positivity (Fig. 5d). When LRA activity was measured by RT-qPCR, significantly higher copies of cell-associated vRNA per μg RNA were observed upon RK-33 and FH-1321 treatment (Fig. 5e and Supplementary Fig. 6b). We also validated that both BIRC5 and HSPA1B mRNA expression was significantly downregulated upon RK-33 treatment, while FH-1321 treatment resulted in decreased expression of BIRC5 (Supplementary Fig. 6c, d). Therefore, treatment with DDX3 inhibitors also demonstrated latency reversal activity and selective

cell death in vRNA-containing CD4+ T cells from HIV-1-infected donors.

**DDX3 inhibition results in a reduction in the size of the inducible viral reservoir**. The induction of the pro-apoptotic AnnexinV marker in HIV-1-expressing cells that we observed upon DDX3 inhibitor treatment was consistent with the notion that DDX3 treatment could result in depletion of the inducible HIV reservoir. We, therefore, determined whether the DDX3 inhibitor-treated, reactivated and pro-apoptotic cells were actually eliminated over time, resulting in a decrease in the size of the inducible viral reservoir in vitro. CD4+ T cells isolated from PLWHIV were cultured in the presence of the DDX3 inhibitors for 5 days and then treated with PMA-Ionomycin to reactivate them to maximal capacity (Fig. 6a). The size of the inducible viral reservoir upon PMA stimulation after culturing with DDX3 inhibitors was measured by three parallel methods of reservoir quantitation: cell-associated vRNA quantification by nested RT-qPCR, FISH-flow and TILDA (Fig. 6a). We confirmed that culturing patient CD4+ T cells with DDX3 inhibitors for 5 days did not significantly impair viability (Supplementary Fig. 6e), consistent with the mild change in overall gene expression observed by RNA sequencing. Strikingly, as determined by FISH-Flow, CD4+ T cells cultured in presence of the DDX3 inhibitors RK-33 and FH-1321 showed a 35 (s.d. ± 24.9)% and 47.8 (s.d. ± 20.35)% decrease in the frequency of vRNA-expressing cells respectively as compared to the mock-treated cells (Fig. 6b, c and Supplementary Fig. 6f). Similarly, upon treatment with PMA-Ionomycin, a significant decrease in cell-associated vRNA production was observed by nested RT-qPCR in the RK-33 and FH-1321-treated cells as compared to the Mock-treated cells (Fig. 6d and Supplementary Fig. 6g). For three representative donors, we also conducted TILDA to quantify the inducible viral reservoir and observed a decrease in the size of the viral reservoir after 5 days of treatment with RK-33 (Fig. 6e–g and Supplementary Fig. 6h). To confirm that RK-33 and FH-1321 do not impair PMA-induced HIV-1 reactivation, we examined the effect of co-treatment with PMA and DDX3 inhibitors in the Lassen and Greene in vitro CD4+ T cell model of HIV-1 latency. Co-treatment of cells with PMA together with RK-33 or FH-1321 did not impair PMA-induced reactivation, confirming that the decrease in reactivation observed is indeed due to a reduced number of latently infected cells (Supplementary Fig. 6i). Altogether, our data demonstrate that not only do DDX3 inhibitors function as LRAs in primary CD4+ T cells from HIV-1-infected donors, they also selectively ICD in the vRNA-expressing cells, resulting in their elimination and a decrease in the size of the inducible viral reservoir (Fig. 7).

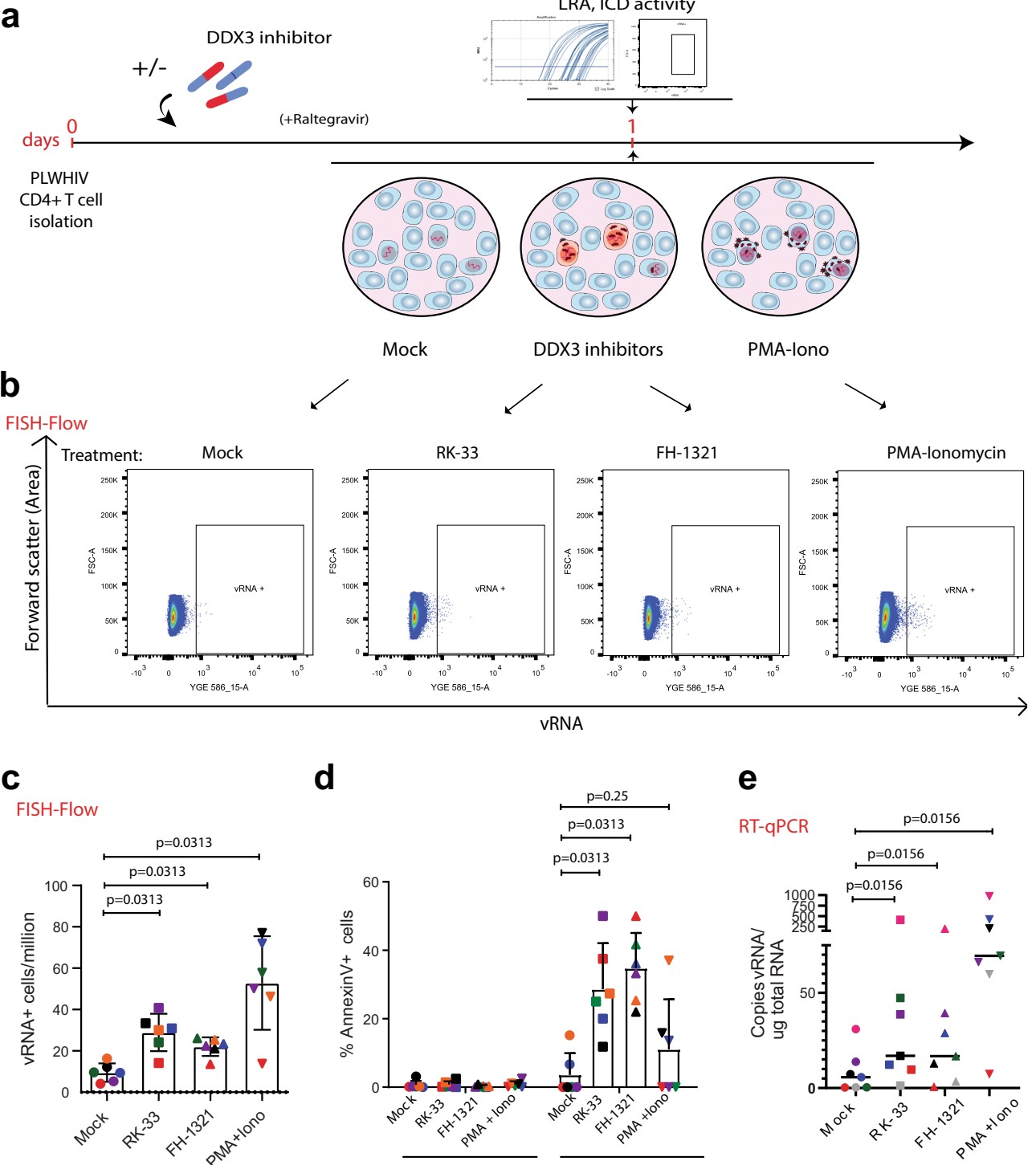

**Fig. 5 DDX3 inhibition reverses latency in CD4+ T cells from HIV-1-infected donors and selectively induces cell death. a** Schematic of the experiment to test the LRA and ICD activity of DDX3 inhibitors in samples from PLWHIV. CD4+ T cells isolated from HIV-1-infected donors were treated with DMSO (unstimulated), PMA/Ionomycin, and the DDX3 inhibitors RK-33 and FH-1321. For all graphs, different donors are represented by their individual colours consistently as described in Table 3 and symbols represent treatments (Mock = circles, RK-33 = squares, FH-1321 = upward triangles and PMA = downward triangles). **b** Representative FISH-Flow dot plots from an HIV-1-infected donor as treated in **a** representing 100,000 cells from each condition. Gating strategy depicted in Supplementary Fig. 5a. **c** The frequency of vRNA-expressing cells per million as depicted in **b** were quantified. Error bars represent the mean ± SD of independent experiments with cells from six different donors, with a minimum of 100,000 events acquired for each condition (Wilcoxon test). **d** The percentages of AnnexinV positive cells from the vRNA− or the vRNA+ cells as quantified by FISH-Flow. Error bars represent mean ± SD from independent experiments from six different donors, with a minimum of 100,000 events acquired for each condition (Wilcoxon test). **e** Copies of vRNA/μg RNA from cells as treated in **a** were quantified by nested RT-qPCR. Error bars represent the mean ± SD of independent experiments with cells from seven different donors (Wilcoxon test).

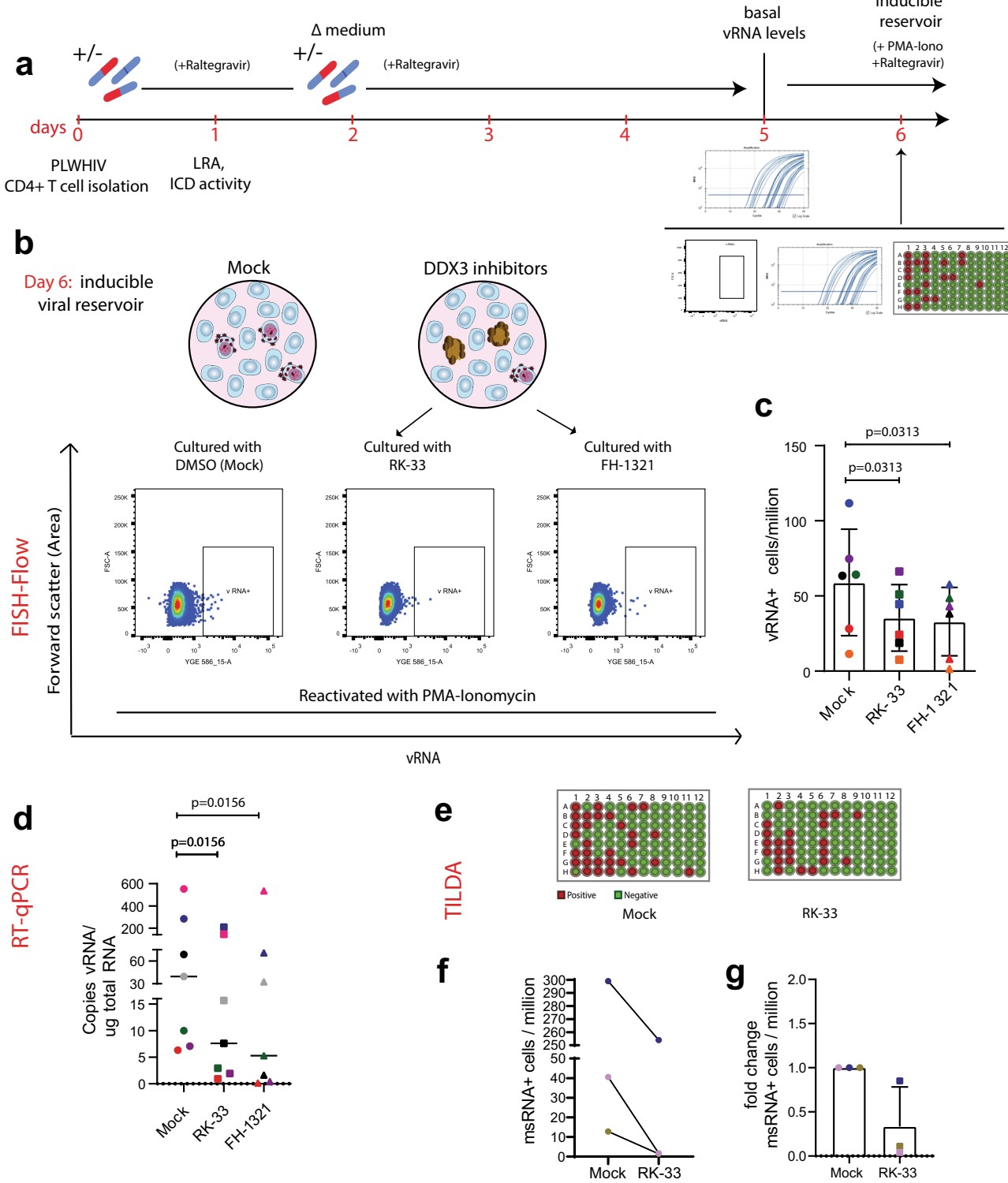

## Discussion

We demonstrate that the treatment of latent HIV-1-infected cells with pharmacological inhibitors of the host RNA-binding protein DDX3 modestly reverses viral latency and impairs vRNA translation. This leads to the accumulation of vRNA in infected cells, which activates the innate antiviral signalling pathways[17,18,62,63], phosphorylates IRF3 and induces the production of IFN-β that renders vRNA+ cells pro-apoptotic (Fig. 2e–k)[50–52]. In combination with the DDX3 inhibition-induced downregulation of the anti-apoptotic proteins BIRC5 and HSPA1B, this results in the

selective induction of apoptosis in HIV-1 vRNA-expressing cells, with minimal toxicity to uninfected/bystander cells that do not express pro-apoptotic IFN-β (Fig. 7). Importantly, we demonstrate a reduction in the size of the inducible viral reservoir when latently infected cells were cultured in the presence of DDX3 inhibitors. Our data support the proof of concept that pharmacological induction of selective cell death in HIV-1 infected cells following latency reversal may be a viable HIV-1 curative strategy.

The ability of pro-apoptotic molecules to induce NF-κB pathways and reactivate HIV-1 latency is not without

**Fig. 6 Prolonged DDX3 inhibition can reduce the size of the inducible viral reservoir. a** Schematic of the experiment to test the size of the inducible viral reservoir after culturing CD4+ T cells from PLWHIV with DDX3 inhibitors. CD4+ T cells isolated from HIV-1-infected donors were cultured in the presence of the DDX3 inhibitors RK-33 and FH-1321 or mock-treated for 5 days. On day 5, basal vRNA levels were measured and then cells were maximally stimulated with PMA-Ionomycin for 12–18 h. On day 6, the inducible viral reservoir was measured by FISH-Flow, vRNA RT-qPCR and TILDA. For all graphs, different donors are represented by their individual colours consistently as described in Table 3 and symbols represent treatments (Mock = circles, RK-33 = squares and FH-1321 = triangles). **b** Representative FISH-Flow dot plots on day 6 of cultured cells from an HIV-1-infected donor as treated in **a** representing 50,000 cells from each condition. Gating strategy depicted in Supplementary Fig. 5a. **c** The frequency of vRNA+ cells/million on day 6 as depicted in **b** were quantified and normalised to the unstimulated condition. Error bars represent the mean ± SD of independent experiments with cells from six different donors in duplicate, with a minimum of 100,000 events acquired for each condition (Wilcoxon test). **d** Copies of vRNA/μg total RNA quantified by RT-qPCR on day 6 as treated in **a** and depicted in **d**. Error bars represent the mean ± SD of independent experiments with cells from seven different donors (Wilcoxon test). **e** Representative TILDA results from cells treated with DDX3 inhibitors. **f** Frequency of msRNA-expressing cells as measured by TILDA and (**g**) fold change of the frequency of msRNA-expressing cells upon treatment with RK-33 normalised to the Mock-treated cells. Error bars represent the mean ± SD of independent experiments with cells from three donors.

precedence. Second mitochondria-derived activator of caspase (SMAC) mimetics such as LCL-161, SBI-0637142, birinapant, Debio1143 and AZD5582 that target the host anti-apoptotic factors XIAP and cIAP1/BIRC2 have been demonstrated to be potent LRAs via the activation of the non-canonical NF-κB pathways[64–66]. Some LRAs have also been demonstrated to promote cell death, as seen with benzolactam derivatives that are PKC agonists and induce apoptosis in latent HIV-1-infected cells[67,68]. A recent report also demonstrated that romidepsin treatment in a clinical trial resulted in an increase in the frequency of apoptotic T cells[69]. Therefore, the link between apoptosis and latency reversal is highly promising to further exploit as a strategy to reactivate the viral reservoir and induce its selective elimination. The DDX3 inhibitors described in this report are such compounds that have both LRA and ICD capability and were able to decrease the size of the inducible viral reservoir.

DDX3 inhibitors are especially interesting as a potential therapeutic class of compounds for use in curative strategies against HIV-1 because they target multiple steps of the HIV-1 life cycle. In this report, we describe the effects of DDX3 inhibition on viral reactivation, possibly mediated by the activation of NF-κB[46] and in agreement with a previous study, we found DDX3 inhibitor-mediated hindering of vRNA export[32,35]. Additionally, DDX3 inhibitors may also impair the translation of the vRNA[34,36]. Recent studies have demonstrated that the treatment of PBMCs with DDX3 inhibitors targeting the RNA-binding site results in a decrease in released infectious viral particles and that these compounds are also effective against HIV-1 strains harbouring resistant mutations to currently approved antiretroviral drugs[38,70]. This characteristic impairment of host cell translation is an advantage when applying DDX3 inhibitors towards an HIV-1 cure strategy since the generation of infectious viral particles is impaired and therefore treatment with DDX3 inhibitors would prevent both the unnecessary activation of the cell-mediated and humoral immune responses or inflammation induced by viral production. Additionally, we demonstrate a role for DDX3 inhibitors as ICD inducers to eliminate the viral reservoir. DDX3 inhibitors demonstrated the ability to target the HIV-1-infected cells selectively, an important characteristic when developing ICD inducers. In this work, we demonstrate that the DDX3 inhibitors have negligible effects on the viability of uninfected cells as well as on non-HIV-1 vRNA-containing cells from PLWHIV at concentrations effective in reversing latency and selectively inducing apoptosis in cells harbouring HIV-1. We hypothesise that this selectivity may arise from the contribution of several factors. Firstly, DDX3 inhibition results latency reversal, possibly through the activation of NF-κB (Fig. 1). The vRNA generated upon latency reversal activates innate antiviral signalling pathways resulting in the phosphorylation of IRF3 and the production of

IFN-β, thereby rendering vRNA-expressing cells pro-apoptotic (Fig. 2e–k). Therefore, vRNA-producing cells are more susceptible to dying that uninfected cells that do not express IFN-β (Fig. 2i). DDX3 inhibition could also abrogate its function as a pathogen recognition receptor and instead could lead to increased vRNA levels for sensing by other factors, including the RIG-I like receptors, thereby enhancing innate antiviral signalling. The second factor that contributes to the selectivity of cell death is that DDX3 inhibition downregulates the expression of the anti-apoptotic protein BIRC5 in both healthy and HIV-1-infected cells. Since the infected cells are already pro-apoptotic due to IFN-β production, they are selectively eliminated leaving the uninfected/bystander cells unharmed. Consistent with our observations, BIRC5 is involved in the survival of latent HIV-1-infected cells and treatment with BIRC5 inhibitors resulted in cell death of HIV-1 protein-expressing cells[54]. Our data are thus consistent with a model whereby BIRC5 is essential to protect cells from the additional stress induced by the presence of HIV-1. HSPA1B that codes for Hsp70 is another gene that we found significantly downregulated in CD4+ T cells in response to treatment with the DDX3 inhibitor RK-33. Hsp70 has been shown to protect cells from apoptosis induced by the HIV-1 protein Vpr[55]. Therefore, we can speculate that an additional mechanism of RK-33 by which it promotes HIV-1 selective cell death is through downregulation of Hsp70, whose absence leads to induction of cell death in the Vpr-expressing infected cells and not in the uninfected bystander cells. Additionally, the depletion of Hsp70 via binding to HSPBP1 has been implicated in the reactivation of HIV-1 via NF-κB activation[57], providing another possible mechanism for the observed latency reversal property of RK-33. Moreover, DDX3 inhibition interferes with vRNA metabolism and the residual vRNA that does not efficiently generate infectious viral particles could still induce cellular stress[18], activate innate antiviral signalling pathways and induce IFN-β production[17] and, in combination with the downregulation of factors important for the survival of latently infected cells, resulting in the selective killing of HIV-1-infected cells. This selective toxicity is also important when promoting the translation of these compounds into clinical settings. The paracrine effect of the production of Type I IFNs on uninfected/bystander cells as previously observed in the context of HIV-1 infection[62,63] was negligible in our experiments, likely because of the very low percentages of vRNA-producing primary CD4+ T cells in both our in vitro model system and from PLWHIV donors (Figs. 4c and 5c). We validated that both RK-33 and FH-1321 have a negligible effect on cell viability and gene expression in primary cells and that they do not impair the effector function and proliferation of CD4+ T cells. Importantly, these compounds also do not affect CD8+ T cell effector function and proliferation so that the essential cell-mediated immune responses would not be

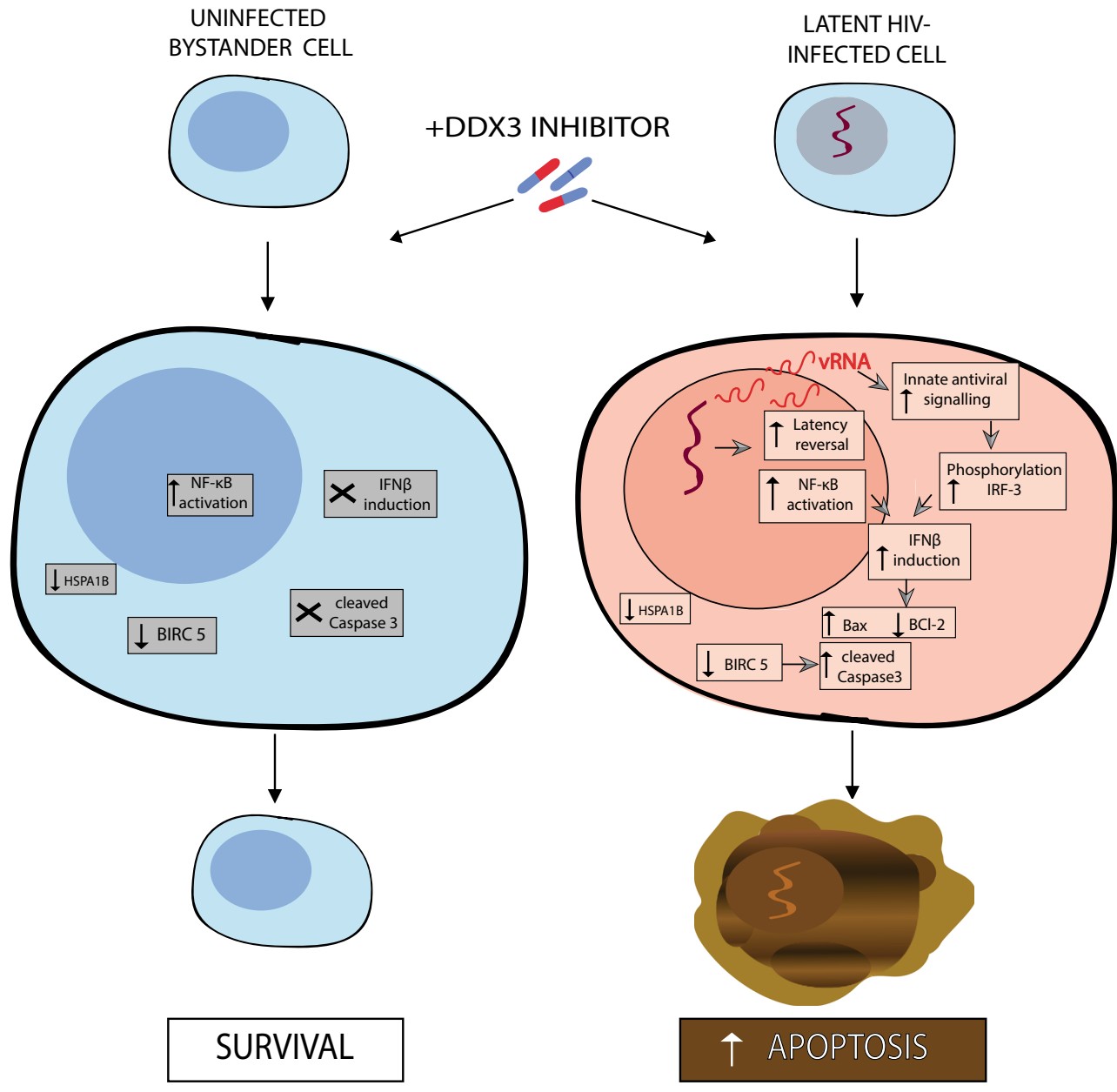

### SELECTIVE KILLING OF VIRAL RESERVOIR

**Fig. 7 Model for latency reversal and cell death induced by DDX3 inhibition.** DDX3 inhibition results in HIV-1 latency reversal, possibly via NF-κB activation, and BIRC5 and HSPA1B downregulation in both uninfected bystander and latent proviral-containing cells. The HIV-1 provirus is reactivated generating vRNA that can activate innate antiviral signalling pathways, phosphorylate IRF3 and induce IFNβ expression, which when combined with BIRC5 or HSPA1B downregulation, results in upregulated Bax, downregulated Bcl2, increased cleaved caspase-3 and apoptosis of the vRNA-expressing cells, but leaving the uninfected cells unharmed, thereby resulting in a reduction in the viral reservoir.

affected during treatment. Extensive toxicology, biodistribution and pharmacokinetics of RK-33[40] and FH-1321[38] have been conducted in mice and rats respectively, and both demonstrate a good toxicology profile in these pre-clinical studies, thereby advocating for their translation into clinical studies. DDX3 inhibition has however been shown to alter the cell cycle by causing G1 cell cycle arrest, a factor not evaluated in the context of this work, but to be noted for future work or clinical translation. Next-generation DDX3 inhibitors with significantly lower

IC50s have also recently been developed with a high potential for clinical advancement[70].

We also present in this work a primary CD4+ T cell model that we have established as a pipeline to further investigate compounds that ICD and to test their abilities to reduce the size of the inducible viral reservoir using three parallel methods of reservoir quantitation (Fig. 5a). Since the ICD strategy does not depend on other components of the host immune system such as CD8+ T cells or NK cells, this system can be used to measure

pro-apoptotic compounds that have previously been studied to selectively eliminate HIV-1-expressing cells. Some examples of compounds that can be tested include Bcl2 agonists such as venetoclax[71,72]; SMAC mimetics like birinipant, GDC-0152 and embelin[73]; Benzolactam derivatives such as BL-V8-310[67]; p38/ JNK pathways activator anisomycin[18], specific autophagy-inducing peptides Tat-Beclin 1 and Tat-vFLIP-α2[74]; PI3K/Akt inhibitors[75]; or synthesised compounds such as L-HIPPO that inhibit viral trafficking and budding of Gag[76] that have all been implicated in killing of HIV-1 infected cells. Pharmacological inhibition of BIRC5, also one of the genes downregulated by DDX3 expression, has been demonstrated to target HIV-1-infected cells and reduce the frequency of proviral-containing cells in vitro[54]. However, some of these treatments have been associated with global T cell activation and toxicity, selective myeloid cell-type-dependent activity or non-reproducibility. Moreover, these studies have not specifically investigated the contribution of the HIV-1 vRNA in inducing cell death, made possible here using FISH-Flow technology. Further investigation of these compounds, possibly in combination with DDX3 inhibitors, will be important to assess their ability to decrease the size of the inducible viral reservoir in CD4+ T cells obtained from PLWHIV using an in vitro model such as the one we describe in Fig. 6a, and also to investigate the contribution of the vRNA to cell death. These studies suggest that pharmacological induction of selective cell death in HIV-1-infected cells, alone or in combination with other approaches, including those that promote extracellular killing of HIV-1-infected cells may prove effective towards an HIV-1 cure.

One of the caveats of this study is that DDX3 inhibitors are modest LRAs that do not maximally reactivate the HIV-1 provirus as compared to the PMA-treated cells, thereby resulting in reactivation of only a fraction of the latently infected cells and hence a partial decrease of the viral reservoir but not its elimination. Moreover, since our studies have focussed on using RNA-based readouts for the viral reservoir, it is yet to be determined if treatment with DDX3 inhibitors also results in a decrease in the translation competent reservoir. Therefore, a combinatorial strategy will be important to test in future work, combining different classes of LRAs to maximally reactivate the latent HIV-1-infected reservoir together with DDX3 inhibitors with the ultimate aim to ICD in all cells harbouring inducible replication-competent HIV-1, also measuring the amount of released viral particles, thereby accounting for the translational competent reservoir. Attractive candidate LRAs for inclusion in this combinatorial approach are de-repressors of HIV-1 LTR chromatin (BAF and HDAC inhibitors)[60], activators of transcription (PKC agonists), as well as facilitators of HIV-1 transcription elongation such as the newly characterised Gliotoxin[77]. Using a pipeline to measure vRNA production and cell death at a single-cell level may enable the identification of an optimal combination of LRA-ICDs that lead to a maximal reduction of the viral reservoir.

So far more than 160 compounds have been identified with latency-reversing activity, and our work has identified DDX3 inhibitors as another category of compounds to reverse viral latency[78]. In addition to latency reversal, DDX3 inhibitors target multiple steps of the HIV-1 replication cycle: transcription, nucleocytoplasmic export of the vRNA, translation and generation of infectious viral particles. Importantly, they also promoted ICD and were selectively able to induce IFN-β production and apoptosis in vRNA-expressing cells resulting in the decrease in size of the inducible viral reservoir. Previous combinatorial studies and clinical trials with LRAs and immune-enhancing clearance strategies demonstrated minimal effects on the clearance of the viral reservoir[4–6]. Although multiple compounds have been identified to induce apoptosis of HIV-1-infected cells[13], no ICD inducer has been evaluated for the capability to reduce the viral reservoir in PLWHIV yet. Our data support DDX3 inhibitors as a promising class of ICD inducers that can be utilised in HIV-1 curative therapies.

## Methods

**Cell culture and reagents.** Jurkat cells and J-Lat 11.1 cells (Jurkat cells containing integrated full-length HIV-1 genome mutated in *env* gene and GFP replacing *nef*)[39] were maintained in RPMI-1640 medium (Sigma-Aldrich) supplemented with 7% FBS and 100 μg/ml penicillin-streptomycin at 37 °C in a humidified, 5% $CO_2$ atmosphere. J-Lat latent proviruses were reactivated by adding 10 μM of phorbol 12-myristate 13-acetate (PMA) (Sigma-Aldrich) to the culture media for 18 h. Primary CD4+ T cells were isolated from buffy coats from healthy donors by Ficoll gradient followed by separation via negative selection with RosetteSep Human CD4 + T-cell Enrichment Kit (StemCells Technologies) according to the manufacturer's instructions. After isolation, CD4+ T cells were maintained in RPMI-1640 medium (Sigma-Aldrich) supplemented with 10% FBS and 100 μg/ml penicillin-streptomycin at 37 °C in a humidified, 5% $CO_2$ atmosphere. Cells were activated with 100 ng/ml PMA and 1 μg/ml Ionomycin (Sigma-Aldrich). HEK293T cells were cultured in DMEM medium (Sigma-Aldrich) supplemented with 10% FBS and 100 μg/ml penicillin-streptomycin at 37 °C in a humidified, 5% $CO_2$ atmosphere. RK-33 (SelleckChem) was used at a concentration of 2 μM unless indicated otherwise. FH-1321 was provided by First Health Pharmaceuticals and used at concentrations of 1 μM. YM155 (SelleckChem) was used at concentrations of 200 nM.

**Total RNA isolation and quantitative RT-PCR (RT-qPCR).** Total RNA was isolated from the cells using TRIzol reagent (Thermo Fisher Scientific) according to the manufacturer's instructions. cDNA synthesis was performed using Superscript II Reverse Transcriptase (Life Technologies) kit following the manufacturer's protocol. RT-qPCR was performed using GoTaq qPCR Master Mix (Promega) following the manufacturer's protocol. Amplification was performed on the CFX Connect Real-Time PCR Detection System thermocycler using CFX Manager version 3.0 (Bio-Rad) using the following thermal cycling programme starting with 3 min at 95 °C, followed by 40 cycles of 95 °C for 10 s and 60 °C for 30 s. The specificity of the RT-qPCR products was assessed by melting curve analysis. Primers used for real-time PCR are listed in Supplementary Table 1. Expression data were calculated using a $2^{-\Delta\Delta Ct}$ method[79]. GAPDH housekeeping gene expression was used for normalisation for J-Lat cell lines and Cyclophilin A was used for primary cells.

**NF-κB reporter assay.** To generate the plasmids used in this assay, custom gBlock gene fragments (Integrated DNA Technologies) containing either the minimal mouse heat shock protein (Hspa1a) promoter, or four NF-κB target sites followed by an Hspa1a promoter were designed with flanking NheI-NcoI restriction sites and then cloned upstream of the luciferase gene in the pGL3 luciferase reporter plasmid at the NheI-NcoI sites (Promega). Jurkat cells were transfected with pBluescript II SK(+) (empty vector), pRL Renilla Luciferase Control Reporter Vector (Promega) to normalise for transfection efficiency, and either the HSP-Luc or the NF-κB-HSP-Luc plasmids by nucleofection using Amaxa Nucleofector (Amaxa AG-Lonza, Cologne, Germany)[59]. Ten million cells were centrifuged at $200 \times g$ for 10 min at room temperature, resuspended in 100 μl of solution R, and nucleofected with 2 μg plasmid DNA (1.5 μg pBluescript II SK(+), 200 ng pRL and either Hspa1a-pGL3 or NF-κB-Hsp1a-pGL3) using programme O28. Nucleofected cells were resuspended in 5 ml of pre-warmed, serum-free antibiotic-free RPMI at 37 °C for 15 min and then plated in 6 ml of pre-warmed complete media. Twenty-four hours post-nucleofection cells were either mock-treated or treated with 2 μM RK-33. Luciferase expression was measured 24 h post-treatment in a GloMax 96 microplate luminometer using the GLOMAX software (version 1.9.2) (Promega) using the Dual-Glo Luciferase assay system (Promega) and normalised to Renilla luciferase expression. Data represent at least three independent experiments.

**FISH-Flow.** Cells were collected, fixed, permeabilised and subjected to the PrimeFlow RNA assay (Thermo Fisher Scientific) following the manufacturer's instructions and as described in[20,21]. In primary CD4+ T cells, cells were first stained in Fixable Viability dye 780 (Thermo Fisher Scientific) for 20 min at room temperature (1:1000 in dPBS) followed by either AnnexinV-450 (560506, BD Biosciences) or with 2 μM CellEvent™ Caspase-3/7 Green Detection Reagent (Thermo Fisher Scientific) for 30 min at room temperature (1:150 in AnnexinV binding buffer-Biolegend). For both CD4+ T cells and J-Lat 11.1 cells, mRNA was labelled with a set of 40 probe pairs against the GagPol region of the vRNA (catalogue number GagPol HIV-1 VF10-10884, Thermo Fisher Scientific) diluted 1:5 in diluent provided in the kit and hybridised to the target mRNA for 2 h at 40 °C. Positive control probes against the housekeeping gene RPL13A (VA1-13100, Thermo Fisher Scientific) were used to determine assay efficiency. Samples were washed to remove excess probes and stored overnight in the presence of RNAsin. Signal amplification was then performed by sequential 1.5 h, 40 °C incubations with the pre-amplification and amplification mix. Amplified mRNA was labelled with

fluorescently tagged probes for 1 h at 40 °C. Samples were acquired on a BD LSR Fortessa Analyzer and gates were set using the unstimulated J-Lat 11.1 control sample or uninfected CD4+ T cells. The analysis was performed using the FlowJo V10 software (Treestar).

**Confocal microscopy following FISH-Flow**. J-Lat 11.1 cells that underwent the FISH-Flow assay were seeded on 18 mm diameter coverslips and air-dried. Coverslips were mounted in ProLong Gold Antifade Reagent (Life Technologies). Immunofluorescence images were acquired using a confocal microscope (Leica, SP5). All phase-contrast pictures were acquired using a Leica DMIL microscope and a DFC420C camera. Images were analysed and processed using Fiji/ ImageJ (NIH).

**Production of shRNA lentiviral vectors and transduction**. Lentiviral constructs containing the desired shRNA sequences (shControl—SHC002 and shDDX3) were amplified from bacterial glycerol stocks obtained in house from the Erasmus Center for Biomics and part of the MISSION® shRNA library. In total, $5.0 \times 10^6$ HEK293T cells were plated in a 10-cm dish and transfected with 12.5 µg of plasmids mix. In total, 4.5 µg of pCMVΔR8.9 (envelope)[80], 2 µg of pCMV-VSV-G (packaging)[80] and 6 µg of shRNA vector were mixed in 500 µl serum-free DMEM and combined with 500 µl DMEM containing 125 µl of 10 mM polyethyleneimine (PEI, Sigma). The resulting 1 ml mixture was added to HEK293T cells after 15 min incubation at room temperature. The transfection medium was removed after 12 h and replaced with a fresh RPMI medium. Virus-containing medium was harvested and replaced with fresh medium at 36, 48, 60 and 72 h post-transfection. After each harvest, the collected medium was filtered through a cellulose acetate membrane (0.45 µm pore) and used directly for shRNA infections or stored at −80 °C for subsequent use. For lentiviral transduction of J-Lat 11.1 cells, 1 ml of the collected virus was seeded with $2 \times 10^6$ cells in a final volume of 4 ml. Transduced cells were selected with 1 µg/ml Puromycin 72 h post-infection and cells were collected 4 days post-selection. For primary CD4+ T cells, cells were stimulated for 3 days with anti-CD3-CD28-coated Dynabeads (Thermo Fisher Scientific) and 5 ng/ml IL-2 (Thermo Fisher Scientific), and then resuspended with 1 ml of virus per $6 \times 10^6$ cells. Transduced cells were selected with 1 µg/ml Puromycin 72 h post-infection and cells were harvested 4 days post-selection.

**Western blotting**. At least 500,000 J-Lat 11.1 cells or $2 \times 10^6$ CD4+ T cells were lysed per condition. Cells were washed with dPBS and lysed in ice-cold lysis buffer (100 mM NaCl, 10 mM Tris, pH 7.5, 1 mM EDTA, 0.5% Nonidet P-40, protease and phosphatase inhibitor cocktail [Roche]). Cell lysates were denatured in Laemmli sample buffer and incubated for 5 min at 95 °C. The proteins were separated by SDS-PAGE and transferred onto nitrocellulose membranes (Bio-Rad). Membranes were blocked with 5% nonfat milk in Tris-buffered saline pH 7.4 and 0.5% Tween 20 (TBST) and then incubated with primary antibodies. After three washes with TBST, the membranes were incubated with horseradish peroxidase-conjugated secondary antibodies (Rockland Immunochemicals; 1:10,000) and detected using Western Lightning Plus-ECL reagent (Perkin Elmer). Signal intensity and densitometry analyses were conducted using ImageJ (NIH). The following antibodies were used: anti-Puromycin (12D10 Sigma-Aldrich; 1:1000), anti-GAPDH (ab8245, Abcam; 1:5000) and anti-DDX3 (ab128206, Abcam; 1:2000), anti-Phospho-IRF3 (Ser386) (37829, Cell Signalling Tech; 1:1000), anti-HSP70 (4872, Cell Signalling Tech; 1:1000), anti-IRF3 (sc33641, Santa Cruz; 1:1000), anti BIRC5 (sc17779, Santa Cruz; 1:1000) and anti-tubulin (B7, Santa Cruz; 1:3000).

**Molecular docking**. Molecular docking analysis was used to determine the most likely binding mode of RK-33 and FH-1321 to pre-unwound DDX3 conformation. Crystal structure of human DDX3 in complex with dsRNA (PDB ID code 6O5F) in the pre-unwound confirmation was used as a template to define the receptor for the docking simulation by elimination or protein chain B and RNA domain C and D. RK-33 and FH-1321 ligand structure was built and energy minimised using the programme Chimera[81]. Molecular docking of RK-33 and FH-1321 to pre-unwound DDX3 confirmation was performed using Chimera's AutoDock Vina function[82]. The resulting solutions were ranked based on the highest binding affinity (or lowest binding energy). Figures were created using PyMol Molecular Graphics System, Version 1.8. 2015 (Schrödinger L).

**RNA-seq**. RNA was isolated using TRI Reagent (Sigma-Aldrich). RNA-seq was performed according to the manufacturer's instructions (Illumina) using the TruSeq Stranded mRNA Library Prep kit. The resulting DNA libraries were sequenced according to the Illumina TruSeq v2 protocol on an Illumina HiSeq 2500 sequencer. Reads of 50 bp in length were generated. Reads were mapped against the UCSC genome browser GRCh38 reference genome with HiSat2 (version 2.1.0). Gene expression was quantified using htseq-count (version 0.11.2). For all samples, at least 14.4 million reads were generated with counts on 22.6–24.2 thousand expressed genes. Differential expression analysis of the RNA-seq data was performed using edgeR package run under Galaxy (https://bioinf-galaxian. erasmusmc.nl/galaxy/). False discovery rate cut-off was set to 0.05 and a fold

change of ±1.5 was counted as differentially expressed. Heat maps were generated using MORPHEUS (https://software.broadinstitute.org/morpheus/index.html).

**Viability and activation assays**. In Jurkat and J-Lat cells, 10 ng/ml Hoescht 33342 was added to the cells and 30 min later, the viability and reactivation were quantified by measuring 405 nm+ and GFP+ expression, respectively, by flow cytometry. In primary CD4+ T cells, to determine viability and cell activation following DDX3 inhibition, cells were collected and stained with Fixable Viability Dye eFluor® 780 (Thermo Fisher Scientific) and anti-CD25-PE (Becton Dickinson). Mock-treated and cells treated with PMA-Ionomycin were used as negative and positive controls, respectively. Cells were stained for 30 min at 4 °C, washed with PBS and resuspended for flow cytometric analysis.

**T cell proliferation and functionality assay**. To analyse the effect of the LRA on CD8+ and CD4+ T cells, proliferation and cytokine expression were analysed by flow cytometry. Primary CD8+ and CD4+ T cells were isolated from buffy coats from healthy donors by Ficoll gradient (GE Healthcare) followed by negative selection with RosetteSep Human CD8+ T-Cell Enrichment Cocktail or RosetteSep Human CD4+ T-Cell Enrichment Cocktail (StemCell Technologies), respectively. Isolated cells were left overnight for recovery. To analyse T cell proliferation capacity, one million CD8+ or CD4+ T cells were stained with 0.07 µM CellTrace Far Red Cell Proliferation dye (Thermo Fisher Scientific) following the manufacturer's instructions. Cells were then cultivated for 72 h in either unstimulated or stimulated conditions in the presence of the LRA, and analysed by flow cytometry. Stimulation of T cells was performed using Anti-CD3/CD28-coated Dynabeads (Thermo Fisher Scientific) following the manufacturer's protocol. Proliferation capacity was determined by a decrease in proliferation of dye intensity in daughter cells upon cell division. To analyse T cell functionality through cytokine expression one million CD8+ or CD4+ T cells were left untreated or treated with the LRA for 18 h. Cells were then left unstimulated or stimulated with 50 ng/ml PMA and 1 µM Ionomycin for 7 h in the presence of a protein transport inhibitor (BD Golgi-Plug™, BD Biosciences). To stain for intracellular cytokines cells were washed with PBS supplemented with 3% FBS followed by fixation and permeabilisation step with FIX & PERM Kit (Invitrogen) following manufacturer's protocol and incubated with 1:25 BV510 anti-IL-2 (563265, BD Biosciences) and PE-Cy7 anti-IFNγ (27-7319-41, eBioscience) in permeabilisation buffer for 45 min at 4 °C. Stained cells were washed with PBS supplemented with 3% FBS and analysed by flow cytometry.

**Measurement of de novo protein synthesis**. De novo protein synthesis upon DDX3 inhibition was measured by the incorporation of puromycin into peptide chains[83]. Briefly, cells were incubated with 10 µg/ml puromycin (Sigma-Aldrich) for 10 min before cell lysis followed by Western blot according to the protocol described above using antibodies against puromycin. Puromycin incorporation was assessed by summating the immunoblot intensity of all de novo synthesised protein bands normalised to the signal intensity of the GAPDH band. CD4+ T cells were treated with 10 µg/ml Cycloheximide (Sigma-Aldrich) for 4 h to inhibit translation as a control for the technique.

**Primary CD4+ T cell infection and in vitro HIV-1 latency model generation**. Infections were performed using a pseudotyped virus expressing luciferase that was generated by co-transfecting HXB2-Env together with the HIV-1 backbone plasmid with a luciferase reporter (pNL4.3.Luc.R-E-) into HEK293T cells using PEI (Polyethylenimine) transfection reagent. In total, 48 and 72 h post-transfection, the pseudovirus-containing supernatant was collected, filtered through a 0.45 µm filter, aliquoted and concentrated by ultracentrifugation ($20,000 \times g$ for 1 h at 4 °C). The pellet was resuspended in RPMI-1640 and stored at −80 °C. The in vitro model of HIV-1 latency was set up using the Lassen and Greene method[58] as follows: CD4+ T cells were infected with the pNL4.3.Luc.R-E- virus by spinoculation (2 h at $1200 \times g$) 24 h after isolation and cultured overnight in RPMI 10% FBS and 100 µg/ ml penicillin-streptomycin in presence of Saquinavir Mesylate (5 µM). Eighteen hours after spin-infection cells were washed and cultured in growth media supplemented with 5 µM Saquinavir Mesylate. Three days after infection cells were treated with DDX3 inhibitors in presence of Raltegravir (3 µM). HIV-1 molecular clone pNL4.3.Luc.R-E-, HIV-1 HXB2-Env expression vector, Saquinavir Mesylate and Raltegravir were kindly provided by the NIH AIDS Reagents. HIV-1 molecular clone pNL4.3.Luc.R-E- and HIV-1 HXB2-Env expression vector were donated by Dr. Nathaniel Landau and Drs. Kathleen Page and Dan Littman, respectively. Cells were harvested 24 h after stimulation with DDX3 inhibitors, washed once in PBS and either lysed for RNA extraction, used for FISH-Flow analysis or lysed in passive lysis buffer to measure luciferase activity using Luciferase Assay System (Promega). Relative light units were normalised to protein content determined by Bradford assay (Bio-Rad).

**Assays with HIV-1-infected donor samples**. PBMCs from HIV-1-infected donors were obtained by Ficoll density gradient isolation of leukapheresis material from donors that were older than 18 years, cART-treated for at least 1 year, with viral loads below 50 copies/ml for more than 2 months. This study was conducted in accordance with and in compliance with the ethical principles of the Declaration of

**Table 3 Characteristics of PLWHIV donors used in this study.**

| Colour code | Sex | Age | Time from HIV-1 diagnosis to leukapheresis | Time on ART at leukapheresis | Time with continuous HIV-1 RNA <50 copies/ml |
|---|---|---|---|---|---|
| Donor 1 = Red | Male | 66 | 52 months | 52 months | 48 months |
| Donor 2 = Dark Green | Male | 75 | 108 months | 81 months | 75 months |
| Donor 3 = Blue | Male | 47 | 85 months | 72 months | 64 months |
| Donor 4 = Black | Male | 67 | 86 months | 86 months | 12 months |
| Donor 5 = Purple | Female | 42 | 55 months | 55 months | 52 months |
| Donor 6 = Orange | Male | 50 | 124 months | 49 months | 48 months |
| Donor 7 = Grey | Male | 35 | 19 months | 19 months | 17 months |
| Donor 8 = Dark pink | Male | 42 | 77 months | 77 months | 71 months |
| Donor 9 = Light pink | Male | 57 | 18 months | 17 months | 2 months |
| Donor 10 = Khaki green | Male | 50 | 100 months | 42 months | 37 months |

Helsinki. HIV-1-infected patient volunteers were informed and provided signed consent to participate in the study. The authors also affirm that research participants provided informed consent for publication of their clinical characteristic presented in Table 3. The study protocol was approved by the Erasmus Medical Centre Medical Ethics Committee (MEC-2012–583). CD4+ T cell isolation was performed using negative selection with the EasySep Human CD4+ T-cell Enrichment kit (StemCells Technologies) according to the manufacturer's instructions. After isolation, CD4+ T cells were allowed to rest for at least 4 h, maintained in RPMI-1640 medium (Sigma-Aldrich) supplemented with 10% FBS, 100 µg/ml penicillin-streptomycin and Raltegravir (3 µM) at 37 °C, in a humidified, 5% $CO_2$ atmosphere. Cells were treated with DDX3 inhibitors or with PMA-Ionomycin as a positive control for viral reactivation (100 ng/ml PMA and 1 µg/ml Ionomycin). Eighteen hours post-treatment, cells were either lysed in TRIzol for RNA extraction and RT-qPCR or analysed by FISH-Flow. To measure the depletion in the size of the inducible viral reservoir, cells were cultured for 5 days in 2 µM RK-33 and 1 µM FH-1321 with a change in media and retreatment after 48 h. Five days post-treatment, media was changed, and cells were treated with PMA-Ionomycin for 12 h and collected for TILDA, or after 18 h for RT-qPCR and FISH-Flow.

**Cell-associated vRNA measurement by nested RT-qPCR**. Cell-associated vRNA[84] was measured by performing a first round of the PCR using the primers Gag1 (5′ TCAGCCCAGAAGTAATACCCATGT 3′) and SK431 (5′ TGCTATGTC AGTTCCCCTTGGTTCTCT 3′) that amplifies a region within the HIV-1 gag gene in 25 µl of PCR mix containing 4 µl of cDNA, 2.5 µl 10X Platinum Taq buffer, 2 mM MgCl2, 0.4 mM concentrations of deoxynucleoside triphosphates, 0.2 µl Platinum Taq and 0.3 µM each of both primers. The PCR settings were as follows: 95 °C for 5 min, followed by 15 cycles of 95 °C for 30 s, 55 °C for 30 s and 72 °C for 1 min. The product of the first PCR was subsequently used as a template in the second, semi-nested, real-time, PCR amplification. A total of 2 µl of the first PCR product was added to a PCR mix with a final volume of 25 µl containing 2.5 µl 10X Platinum Taq buffer, 2 mM MgCl2, 0.4 mM concentrations of deoxynucleoside triphosphates, 0.2 µl Platinum Taq and 0.2 µM of the primers Gag1 (5′ TCAGCC CAGAAGTAATACCCATGT 3′) and Gag2 (5′ CACTGTGTTTAGCATGGTG TTT 3′) and 0.075 µM TaqMan dual-labelled fluorescent probe Gag3 (5′ [6FAM] ATTATCAGAAGGAGCCACCCCACAAGA[BHQ1] 3′). Real-time PCR settings were as follows: 95 °C for 5 min, followed by 45 cycles of 95 °C for 30s and 60 °C for 1 min. The amplicon sizes were 221 bp for the first PCR and 83 bp for the second (real-time) PCR. Serial dilutions of plasmid DNA standards were used to generate a standard curve to calculate the copies of vRNA and then normalised to the amount of input RNA.

**Tat/rev-induced limiting dilution assay (TILDA)**. TILDA was performed using a modified method as described in[85]. Briefly, total CD4+ T cells were isolated from PBMCs by negative magnetic separation using EasySep Human CD4+ T-cell isolation kit (StemCell Technology). Following treatment for 5 days (Fig. 6a), $2 \times 10^6$ cells/ml stimulated for 12 h with 100 ng/ml phorbol 12-myristate 13-acetate (PMA) and 1 µg /ml ionomycin (both from Sigma-Aldrich). After stimulation, CD4+ T cells were washed and resuspended in RPMI-1640 at serial dilutions; $1.8 \times 10^6$ cells/ml, $9 \times 10^5$ cells/ml, $3 \times 10^5$ cells/ml and $1 \times 10^5$ cells/ml. In total, 10 µl of the cell suspension from each dilution was dispensed into 24 wells of a 96-well plate containing 2 µl One-step RT-PCR enzyme (Qiagen), 10 µl 5x One-step RT-PCR buffer (Qiagen), 10 µl Triton X-100 (0.3%), 0.25 µl RNAsin (40 U/µl), 2 µl dNTPs (10 mM each), 1 µl of tat 1.4 forward primer and rev reverse primer (20 µM) (as published in[86]), and nuclease-free water to a final reaction volume of 50 µl. The one-step RT-PCR was run using the following thermocycling conditions: 50 °C for 30 min, 95 °C for 15 min, 25 cycles of 95 °C for 1 min, 55 °C for 1 min and 72 °C for 2 min, and a final extension at 72 °C for 5 min. Afterwards, 2 µl of the 1st PCR products were used as input for real-time PCR to detect tat/rev msRNA. The 20 µl reaction volume consisted of 5 µl Taqman Fast Advanced Master mix 4x (Thermo Fisher Scientific), 0.4 µl of tat 2.0 forward primer and rev reverse primer (each at 20 µM) and tat/rev probe (5 µM)[86]. The real-time PCR was performed in a LightCycler 480 Instrument II (Roche) using the following programme: 50 °C for 5 min, 95 °C for 20 s, 45 cycles of 95 °C for 3 s and 60 °C for 30 s. Positive wells at each dilution were recorded and used to determine the frequency of cells expressing tat/rev msRNA by maximum likelihood method using ELDA software[87].

**Statistical analysis**. All data are means ± SD of three or more independent biological replicates, as indicated in the figure legends. Statistical significance was calculated using a two-tailed t-test or as indicated in the figure legends. Analyses were performed using Prism version 8.3.0 (GraphPad software).

**Reporting summary**. Further information on research design is available in the Nature Research Reporting Summary linked to this article.

## Data availability

All data needed to evaluate the conclusions in the paper are present in the paper and/or the Supplementary materials. All RNA-seq data have been deposited to the Gene Expression Omnibus (GEO) database with accession code GSE167553. Additional data related to this paper may be requested from the authors. Source data are provided with this paper.

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

## Acknowledgements

This work is dedicated to the memory of C.A.B.B. We would like to thank Jan-Willem Bakker, Alessia Tarditi and Matteo Andreini from First Health Pharmaceuticals Amsterdam for the generous provision of the DDX3 inhibitor FH-1321; and Tsung Wai Kan for technical help. The research leading to these results has received funding from the Dutch Aidsfonds (grants P-53302 and P-53601), Health Holland (grants LSHM19100-SGF and EMCLSH19023), ZonMW (grant 40-44600-98-333), the Federation of Medical Specialists (grant 59825822) and the EHVA T01 consortium, which is supported by the European Union's Horizon 2020 Research and Innovation Programme (grant 681032).

## Author contributions

S.R. and T.M. designed the studies; S.R., C.L., R.C., T.H.S. and A.G. conducted experiments and performed data analysis; S.R., R.J.P. and W.V.I. participated in gene expression analysis; H.A.B.P., C.R. and A.V. recruited patients for the study; Y.M.M., P.D.K., J.J.A.V.K., C.A.B.B., R.A.G. and T.M. provided expertise and supervision; S.R. and T.M. wrote the manuscript with input from all authors. All authors read and approved the final manuscript.

## Competing interests

The authors declare no competing interests.
