## [Peer Review File · Nature Communications]

REVIEWERS' COMMENTS

Reviewer #1 (Remarks to the Author):

All my concerns have been appropriately addressed.

My only question that needs clarification is in relation to figure 5c-e and figure 6 c,d. What do the black and orange colors represent? What do the two symbols represent? Can this be clarified in the legend?

Reviewer #2 (Remarks to the Author):

Rao and colleagues have added additional findings to their work to address issues arising in the prior review. I believe the data now presented in this initial study suggests an interesting new approach to an old concept (with appropriate citation of prior similar work on this theme), and further work will be needed for definitive proof of this concept. The effects shown in the J-Lat 11.1 cell line appear consistent, although it is still difficult to understand if many of the changes in viral or host gene expression are substantial, as much of the data is normalized and/or expressed in fold-change. Some protein expression data (eg. Fig 2h) is not very convincing, and some effects seem quite variable (eg. Fig. 2G).

There is a major shortcoming that I would urge the authors to consider in their future work. Both basal and induced viral gene expression varies substantially within a cell population in both primary cell models, and especially in cells from PLWH. Further, when viral expression is measured primarily by FISH-flow and other RNA detection methods, the entire transcription-competent viral reservoir is measured, but the translation-competent and true replication-competent reservoirs are not. This might be done by QVOA (the most rigorous but difficult and less sensitive) or PCR of virion RNA

released in to tissue culture media (simpler but less definitive). We recognize that the authors claim that DDX3 inhibition also reduces RNA translation, but as so much of the proviral reservoir expresses viral RNAs that are defective (and some in both the basal and induced states), this larger incompetent reservoir could be more susceptible to clearance by this strategy than the true replication competent reservoir. The depletion of the entire transcription-competent viral reservoir suggested in Fig. 6 does not prove depletion of the true replication competent reservoir.

Reviewer #3 (Remarks to the Author):

Comments to the authors

Authors have made important efforts to reply satisfactorily to most of my previous comments and I would like to congratulate them.

As stated by the authors, this work provides the proof-of-concept for the use of DDX3 inhibitors as LRAs that will lead future research in this topic. Moreover, this work reinforces the idea of the immunogenic capacity of the HIV-1 full-length RNA and the critical role of DDX3 on viral RNA metabolism. In my opinion, the manuscript will be suitable for publication once the following minor comments are addressed:

1) CA HIV-1 RNA and Pol mRNA do not exist... both the CA and the Pol coding regions are contained in the full-length RNA (also called unspliced RNA or 9-kb RNA). Therefore, whatever the methodology employed (RT-PCR, FISH-Flow, etc) and the primers or probes designed to recognize the CA or Pol regions, authors will detect the full-length RNA. Depending on the function it is accomplishing (mRNA or genome), the full-length RNA could be called gag-pol mRNA or genomic RNA. In order to avoid confusions, I suggest the authors to use only one name (e.g., HIV-1 full-length RNA, HIV-1 unspliced RNA or 9-Kb RNA) throughout the manuscript when referring to the viral RNA.

2) Authors should cite recent articles on HIV RNA and immune activation in macrophages, DCs and CD4+ T-cells (PMID: 30150664 and PMID: 30546110)

3) Supplementary Fig. 1c: why the authors are able to detect Gag in uninduced DMSO-treated J-lat cells, which are expected to contain latent proviruses and do not express Gag (at least in amounts allowing detection by Western blot)?

4) From Figure 1, authors conclude that RK-33 exerts its function as an HIV-1 LRA through NF- κ B. However, I still think that there is no compelling evidence supporting this conclusion (e.g., authors

have not performed experiments analyzing the effect of DDX3 inhibition on latency reversal under NF-kB inhibition). I recommend authors to be more cautious in this conclusion.

5) All X axes in the plots should start at 0 and not a 1 (e.g., Fig. 2d, 2g, 2i, 2j, 2k). Authors should also correct the scale of the plot presented in Fig. 2k

6) The fact that the HIV-1 RNA seems to be immunogenic and trigger apoptosis is really interesting... is this because it is not used in the absence of DDX3? However, although there is evidence showing that the HIV RNA is immunogenic (which is very interesting considering that it is transcribed by the host machinery and thus, it is expected to have most of the features of host mRNA), authors have not formally demonstrated that this effect of the viral RNA is occurring in their latency reversal model (e.g., authors have not performed experiments analyzing the effect of DDX3 inhibitors under knockdown of RIG-I and/or other RNA sensor).

7) Results and discussion sections: authors should explain why uninfected/bystander cells are not affected by IFN- β considering the autocrine and paracrine effects of this molecule. Indeed, it was recently shown that the presence of viral RNA in CD4 T-cells, macrophages and dendritic cells resulted in the induction of a pro-inflammatory state that affected bystander cells (PMID: 30150664 and PMID: 30546110)

8) Line 55 and 652: verify wording

9) Line 338: Fig. 3h and 3i are not well cited in the text

Reviewer #4 (Remarks to the Author):

The authors have done a great deal of work to address the comments of the previous review, including text changes and several new experiments. This is much-appreciated. The manuscript is improved, and my previous comments have been fully addressed to satisfaction.

Reviewer #1 (Remarks to the Author):

All my concerns have been appropriately addressed.

We thank the reviewer for their input and for acknowledging that their concerns have been addressed.

My only question that needs clarification is in relation to figure 5c-e and figure 6 c,d. What do the black and orange colors represent? What do the two symbols represent? Can this be clarified in the legend?

The colours and symbols in these figures depict individual donors and treatment conditions respectively. We have now revised the figure legends of figs 5 and 6 to clarify this (lines 497-500 and lines 550-552 of the revised manuscript).

Reviewer #2 (Remarks to the Author):

Rao and colleagues have added additional findings to their work to address issues arising in the prior review. I believe the data now presented in this initial study suggests an interesting new approach to an old concept (with appropriate citation of prior similar work on this theme), and further work will be needed for definitive proof of this concept. The effects shown in the J-Lat 11.1 cell line appear consistent, although it is still difficult to understand if many of the changes in viral or host gene expression are substantial, as much of the data is normalized and/or expressed in fold-change. Some protein expression data (eg. Fig 2h) is not very convincing, and some effects seem quite variable (eg. Fig. 2G).

There is a major shortcoming that I would urge the authors to consider in their future work. Both basal and induced viral gene expression varies substantially within a cell population in both primary cell models, and especially in cells from PLWH. Further, when viral expression is measured primarily by FISH-flow and other RNA detection methods, the entire transcription-competent viral reservoir is measured, but the translation-competent and true replication-competent reservoirs are not. This might be done by QVOA (the most rigorous but difficult and less sensitive) or PCR of virion RNA released in to tissue culture media (simpler but less definitive). We recognize that the authors claim that DDX3 inhibition also reduces RNA translation, but as so much of the proviral reservoir expresses viral RNAs that are defective (and some in both the basal and induced states), this larger incompetent reservoir could be more susceptible to clearance by this strategy than the true replication competent reservoir. The depletion of the entire transcription-competent viral reservoir suggested in Fig. 6 does not prove depletion of the true replication competent reservoir.

We thank the reviewer for acknowledging that we address issues arising in the prior review and that the study presents an interesting new approach to an old concept. We agree with the reviewer that our current study is a proof of concept that measures only the transcriptional competent reservoir and

focuses on the consequences of triggering activation of the reservoir at the transcriptional (not translational level) in inducing specific killing of HIV-1 infected cells. Our future work using this strategy, potentially in combination with other compounds will indeed focus, as the reviewer highlights, on the impact of this strategy in a reduction in the replication competent viral reservoir. We now discuss this in lines 689-696 of the revised manuscript.

Reviewer #3 (Remarks to the Author):

Comments to the authors

Authors have made important efforts to reply satisfactorily to most of my previous comments and I would like to congratulate them.

We thank the reviewer for acknowledging our efforts to address their previous comments.

As stated by the authors, this work provides the proof-of-concept for the use of DDX3 inhibitors as LRAs that will lead future research in this topic. Moreover, this work reinforces the idea of the immunogenic capacity of the HIV-1 full-length RNA and the critical role of DDX3 on viral RNA metabolism. In my opinion, the manuscript will be suitable for publication once the following minor comments are addressed:

1) CA HIV-1 RNA and Pol mRNA do not exist... both the CA and the Pol coding regions are contained in the full-length RNA (also called unspliced RNA or 9-kb RNA). Therefore, whatever the methodology employed (RT-PCR, FISH-Flow, etc) and the primers or probes designed to recognize the CA or Pol regions, authors will detect the full-length RNA. Depending on the function it is accomplishing (mRNA or genome), the full-length RNA could be called gag-pol mRNA or genomic RNA. In order to avoid confusions, I suggest the authors to use only one name (e.g., HIV-1 full-length RNA, HIV-1 unspliced RNA or 9-Kb RNA) throughout the manuscript when referring to the viral RNA.

We thank the reviewer for their comments and realize that we had not clearly defined CA, which could easily be interpreted in this context as referring to Caspid RNA. To clarify this we now define CA-RNA to correspond to cell-associated viral RNA as detected by amplification of the Gag region by nested Gag RT-PCR (Pasternak et. al, 2008; Lines 118 and 476-477 of the revised manuscript). We consistently use the word vRNA to describe the HIV-1 genomic RNA (Line 74 of the revised manuscript). If agreed, we prefer, after the above clarification in the text, to keep this terminology to remain consistent with the way in which these distinct read-outs are referred to in the field.

2) Authors should cite recent articles on HIV RNA and immune activation in macrophages, DCs and CD4+ T-cells (PMID: 30150664 and PMID: 30546110)

We thank the reviewer for this suggestion and have now cited these references (line 580 of the revised manuscript).

3) Supplementary Fig. 1c: why the authors are able to detect Gag in uninduced DMSO-treated J-lat cells, which are expected to contain latent proviruses and do not express Gag (at least in amounts allowing detection by Western blot)?

Although J-Lat cells are a model of HIV-1 latency, approximately 2% of the cells produce vRNA that codes for Gag even in the uninduced, basal condition (Figure 1 d and e). Since 500000 cells are lysed for western blot, these ~ 10000 cells produce a signal strong enough to be detected by Western blot.

4) From Figure 1, authors conclude that RK-33 exerts its function as an HIV-1 LRA through NF-kB. However, I still think that there is no compelling evidence supporting this conclusion (e.g., authors have not performed experiments analyzing the effect of DDX3 inhibition on latency reversal under NF-kB inhibition). I recommend authors to be more cautious in this conclusion.

We observed modest but significant RK-33-induced NF-kB-mediated activation, using a luciferase reporter plasmid containing NF-kB sites in a minimal promoter. This is consistent with the literature where DDX3 was shown to inhibit NF-kB activation by binding to the p65 subunit (Xiang et. al, 2016). However, we agree with the reviewer that the ultimate proof would be to examine latency reversal by these compounds in context of NF-kB inhibition and therefore keep focus on the ICD effect and selective cell-death with DDX3 inhibitors than their modest latency reversal activity throughout the manuscript.

5) All X axes in the plots should start at 0 and not a 1 (e.g., Fig. 2d, 2g, 2i, 2j, 2k). Authors should also correct the scale of the plot presented in Fig. 2k

Based on the recommendations of the reviewer, all X axes plots in figures 2d, 2g, 2i, 2j and 2k now start at 0 and not at 1.

6) The fact that the HIV-1 RNA seems to be immunogenic and trigger apoptosis is really interesting... is this because it is not used in the absence of DDX3? However, although there is evidence showing that the HIV RNA is immunogenic (which is very interesting considering that it is transcribed by the host machinery and thus, it is expected to have most of the features of host mRNA), authors have not formally demonstrated that this effect of the viral RNA is occurring in their latency reversal model (e.g., authors have not performed experiments analyzing the effect of DDX3 inhibitors under knockdown of RIG-I and/or other RNA sensor).

We thank the reviewer for their comments. Indeed the fact that the HIV-1 RNA in conditions of DDX3 depletion appears to be immunogenic is interesting. We intend in our future work to further characterize this observation mechanistically upon RIG-I or MAVS-knockdowns.

7) Results and discussion sections: authors should explain why uninfected/bystander cells are not affected by IFN- β considering the autocrine and paracrine effects of this molecule. Indeed, it was recently shown that the presence of viral RNA in CD4 T-cells, macrophages and dendritic cells resulted in the induction of a pro-inflammatory state that affected bystander cells (PMID: 30150664 and PMID: 30546110)

Thank you for this comment. We think that because of very low percentages of cells that are infected in the primary model system and cells from PLWHIV, any effect on bystander cells is masked and undetectable at the level of pooled RNA but may become visible at the single cell level, for example by RNA-seq. Based on the suggestion of the reviewer, we have now added a line "The effect of the production IFN- β on uninfected / bystander cells was negligible, likely because of the very low percentages of vRNA-producing primary CD4+ T cells in both our in vitro model system and from PLWHIV donors (Fig 4c and 5c)." to the discussion (lines 648 to 650 of the revised manuscript).

8) Line 55 and 652: verify wording

We thank the reviewer for pointing this out and have edited the wording accordingly.

9) Line 338: Fig. 3h and 3i are not well cited in the text

We thank the reviewer for pointing this out and have cited the figures correctly.

Reviewer #4 (Remarks to the Author):

The authors have done a great deal of work to address the comments of the previous review, including text changes and several new experiments. This is much-appreciated. The manuscript is improved, and my previous comments have been fully addressed to satisfaction.

We thank the reviewer for their insight and previous comments, We are delighted that they consider their comments fully addressed and our manuscript improved.